

# WRF-Chem simulations of CO₂ over Western Europe assessed by ground-based measurements

Jiaxin Wang[1,2,3,4], Sieglinde Callewaert[4], Minqiang Zhou[1], Filip Desmet[4], Sébastien Conil[5], Michel Ramonet[6], Pucai Wang[2,3], Martine De Mazière[4]

[1]State Key Laboratory of Atmospheric Environment end Extreme Meteorology, Institute of Atmospheric Physics, Chinese Academy of Sciences, Beijing 100029, China
[2]CNRC & LAGEO, Institute of Atmospheric Physics, Chinese Academy of Sciences, Beijing 100029, China
[3]University of Chinese Academy of Sciences, Beijing 100049, China
[4]Royal Belgian Institute for Space Aeronomy (BIRA-IASB), Brussels 1180, Belgium
[5]ANDRA DISTEC/EES Observatoire Pérenne de l'Environnement, F-55290, Bure, France
[6]Laboratoire des Sciences du Climat et de l'Environnement, LSCE-IPSL (CEA-CNRS-UVSQ), Université Paris-Saclay 91191 Gif-sur-Yvette, France

*Correspondence to*: Sieglinde Callewaert (sieglinde.callewaert@aeronomie.be) and Minqiang Zhou (minqiang.zhou@mail.iap.ac.cn)

**Abstract.** The Weather Research and Forecasting model coupled with Chemistry (WRF-Chem), in its passive tracer option (WRF-GHG), was used to simulate $CO_2$ concentrations over Western Europe during summer 2018. The model performance was evaluated against ground-based observations. Due to the large variety of anthropogenic emissions, we conducted five sensitivity tests using a combination of three different inventories (CAMS-REG-ANT, EDGAR, and TNO) and source-specific vertical emission profiles. Compared with observations from five Integrated Carbon Observation System (ICOS) atmospheric stations, the model captures diurnal $CO_2$ variations at different heights. At the ICOS site in Karlsruhe, Germany, simulated near-surface $CO_2$ mole fractions are highly sensitive to the choice of anthropogenic emission inventory, with discrepancies up to 14.99±31.98 ppm, due to large nearby emission sources. Furthermore, incorporating source-specific vertical profiles notably improves accuracy, increasing the correlation coefficient from 0.53 to 0.78 when using EDGAR. The column-averaged dry-air mole fractions of $CO_2$ ($XCO_2$) from the Total Column Carbon Observing Network (TCCON) are well simulated by WRF-GHG. However, an overestimation of approximately 1.2 ppm was found at the Paris site, likely due to uncertainties in anthropogenic emissions and boundary conditions. In addition, a negative bias was found in early June at most ICOS and TCCON sites, may be attributed to errors in simulated fluxes during the growing season. However, due to the lack of co-located flux observations, the exact cause remains uncertain. Overall, this study demonstrates the capability of WRF-GHG in simulating $CO_2$ over Western Europe, while showing the need for improving model configuration.

## 1 Introduction

Intergovernmental Panel on Climate Change Sixth Assessment Report (IPCC AR6) points out that human-induced climate change has significantly influenced the frequency and intensity of extreme events such as heatwaves, heavy precipitation,



droughts, and tropical cyclones (Pörtner et al., 2022). In recent years, extreme heat events have become increasingly frequent in western Europe, with prolonged durations of heatwaves (Della-Marta et al., 2007; Sousa et al., 2020; Sánchez-Benítez et

al., 2022). The Paris Agreement proposes to limit the global temperature increase to within 2°C above pre-industrial levels. To achieve this long-term goal, governments must implement measures to reduce carbon emissions. The atmospheric mole fractions of carbon dioxide ($CO_2$), a major greenhouse gas (GHG), have steadily risen due to human activities over the last centuries. By March 2025, the global mean mole fractions of $CO_2$ had increased to 426.40 ppm (Lan et al., 2025). Accurate estimation of carbon emissions is a crucial prerequisite for formulating scientifically sound and effective emission reduction

strategies.

The 2019 refinement of the 2006 IPCC Guidelines for National Greenhouse Gas Inventories (Maksyutov et al., 2019) explicitly states that the top-down method, based on atmospheric inverse modelling, can serve as a potential way to support and verify national greenhouse gas inventories. However, several studies identified uncertainties associated with atmospheric transport models as one of the main sources of error in this approach (Díaz Isaac et al., 2014; Feng et al., 2016). Therefore, reducing

transport errors and improving simulation accuracy are essential for improving the reliability of inversion results.

Regional atmospheric transport models have been widely applied to simulate $CO_2$ mole fractions, mainly focusing on national or urban scale (Zhao et al., 2019; Zhao et al., 2023; Thilakan et al., 2024; Yang et al., 2025), but their simulation results still present some drawbacks and limitations. Previous studies have shown that the quality of model simulations is highly dependent on the boundary conditions and emission inventories accuracy (Callewaert et al., 2022; Karbasi et al., 2025). Additionally,

Brunner et al. (2019) pointed out that using $CO_2$ emissions only at the surface causes a significant overestimation of the simulated near-surface $CO_2$ mole fractions, highlighting the need for properly allocating the vertical signature of emissions.

In order to support accurate assessments of greenhouse gas budgets, various ground-based observation networks have been established to provide consistent and high-precision data, such as Integrated Carbon Observation System (ICOS) Atmosphere in Europe. In situ observations provide near-surface $CO_2$ mole fractions data, which can be used to constrain carbon sources

and sinks at local scales. Using the ICOS atmospheric measurements, Ramonet et al. (2020) reported that a severe drought event in Europe in 2018 led to an atmospheric $CO_2$ signal of 1 to 2 ppm at most stations. Ground-based remote sensing observations, on the other hand, offer information on total column abundances. The global Total Carbon Column Observing Network (TCCON, Wunch et al., 2011) provides long-term, high-precision column-averaged mole fraction measurements that commonly serve as validation data for satellite remote sensing observations (Velazco et al., 2019; Yang et al., 2020; Zhou et

al., 2022) and for model verification (Saito et al., 2012; Turner et al., 2015; Ostler et al., 2016). A signal of 0.8 ppm was observed at the Sodankylä TCCON site during the 2018 drought (Ramonet et al., 2020). Since the magnitude of these signals is less than 2 ppm, it is still difficult for satellites to detect such variations (Connor et al., 2016).

Belgium is located in the central part of Western Europe, serving as an important hub connecting France, Germany, the Netherlands, and Luxembourg, and positioned at the intersection of continental transportation. To support climate change

mitigation commitments and policy development, the project Towards a greenhouse gas emission monitoring and VERification system for BElgium (VERBE), led by the Royal Belgian Institute for Space Aeronomy (BIRA-IASB), proposes to establish



an independent, top-down, temporally and spatially resolved Monitoring and Verification Support (MVS) capacity for greenhouse gas (GHG) emissions in Belgium. To our knowledge, there are no ground-based $CO_2$ observations, and there aren't any $CO_2$ model-based studies specifically for the Belgium region currently, while a few studies have been conducted in

neighbouring Western European countries (Lian et al., 2021; van der Woude et al., 2023; Zhao et al., 2023).

The study presented in this paper has been performed in the framework of the VERBE project. It employs the Weather Research and Forecasting Greenhouse Gas model (WRF-GHG; Beck et al., 2011) and multi-source observational data to simulate $CO_2$ mole fractions over Western Europe, with a focus on Belgium and surrounding regions during summer 2018. The aim is to evaluate the model performance in this region and analyse the spatial and temporal variations of $CO_2$ mole fractions. Different

sensitivity tests were used to investigate the impact of different anthropogenic emission inventories on the model simulations performance and to optimize the WRF-GHG configuration, in order to improve the regional greenhouse gas simulation accuracy and carbon budget inversion in the near future.

This paper is structured as follows: the introduction to the WRF-GHG model setup and input datasets is given in Sect. 2. The observational datasets and the statistical metrics used to evaluate the model performance are described in Sect. 3. The

evaluation of the simulation results, focusing on meteorological fields and $CO_2$ concentration fields are presented in Sect. 4. The simulation errors looking especially at anthropogenic emissions and biogenic sources are discussed in Sect. 5. Finally, the conclusions are drawn in Sect. 6.

## 2 WRF-GHG model

The passive tracer option in the WRF model coupled with Chemistry (WRF-Chem), also known as WRF-GHG, can be used

to simulate the spatiotemporal distribution of long-lived GHGs like $CO_2$ and $CH_4$. In this configuration, the long-lived gases are transported in a passive way without any chemical reactions (Beck et al., 2011). Here WRF-Chem model version 4.5.1 was used.



## 2.1 Model settings

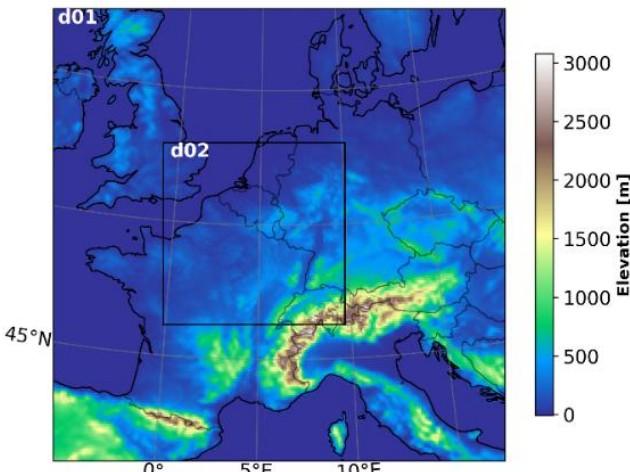

**Figure 1. Location and terrain elevation map of the simulated domains, with horizontal resolutions of 9 km (d01) and 3 km (d02).**

The simulation area covers Western Europe and is centered on Belgium. As shown in Fig. 1 using the Lambert Conformal Conic (LCC) projection, two nested domains are configured, with horizontal resolutions of 9 km for the outer domain and 3 km for the inner domain The vertical grid consists of 60 sigma levels extending from the surface up to 50 hPa. Due to the similarity of the simulation domains, the physical parameterization settings were the same as the settings of Poraicu et al. (2023), except for the urban surface. The Multi-layer Building Effect Parameterization (BEP) model (Martilli et al., 2002) was used instead of the Single-layer Urban Canopy Model (UCM) (Kusaka et al., 2001) because the latter is incompatible with the GHG option. For the Planetary Boundary Layer, we have chosen the Yonsei University scheme (YSU) (Hong et al., 2006).
In the summer of 2018, Europe experienced an intense and widespread heatwave that had a significant impact on ecosystem processes (Bastos et al., 2020; Thompson et al., 2020; Smith et al., 2020). Meanwhile, this period occurred before the outbreak of the COVID-19 pandemic, during which the ground-based observation network operated normally and provided abundant observational data, laying a solid foundation for model validation and analysis. Therefore, our simulation period covers the summer from 1$^{st}$ June to 31$^{st}$ August 2018. In this study, the meteorological fields are re-initialized every 24 hours, with a 6-hour spin-up applied before each re-initialization. By doing this, we can constrain the meteorological fields. This approach has been applied in many studies and has proven to improve the simulations accuracy (Pillai et al., 2011; Zhao et al., 2019; Ho et al., 2024).



**2.2 Input dataset**

In WRF-GHG, the simulated $CO_2$ ($CO_{2,total}$) is the sum of several tracer contributions distinguishing different sources and sinks that are driven by specific emission inventories or flux models, and the so-called background concentrations that are driven by the initial and boundary conditions. Thus, the simulated $CO_2$ concentrations are given by:

$$CO_{2,total} = CO_{2,bck} + CO_{2,ant} + CO_{2,bio} + CO_{2,bbu} + CO_{2,oce} \,, \tag{1}$$

where $CO_{2,bck}$ represents the background concentration, and the remaining terms represent contributions from anthropogenic emissions ($CO_{2,ant}$), biogenic activities ($CO_{2,bio}$), biomass burning emissions ($CO_{2,bbu}$), and ocean-atmosphere exchange ($CO_{2,oce}$), respectively. Table 1 gives an overview of the input datasets employed in the simulation, apart from anthropogenic emissions, along with their temporal and spatial resolutions. For initial and lateral boundary conditions, the meteorological fields are provided by the European Centre for Medium-Range Weather Forecasts (ECMWF) global ERA5 hourly reanalysis dataset on model levels, which includes 137 vertical levels (Hersbach et al., 2020), and the chemical fields are provided by the 3-hourly Copernicus Atmosphere Monitoring Service (CAMS) global greenhouse gas reanalysis (EGG4) which includes 60 vertical model levels (Inness et al., 2019). For the emissions, the open biomass burning flux is obtained from the daily Fire INventory from NCAR (FINN v2.5; Wiedinmyer et al., 2023), and the $CO_2$ fluxes from the oceans are taken from the observation-based global monthly gridded sea surface $pCO_2$ climatology (NCEI; Landschützer et al., 2017). Given the significant influence of anthropogenic activities throughout much of the study area, the choice of anthropogenic emission inventory can greatly affect the simulation accuracy. The sensitivity tests using different inventories and setups will be discussed later (see Sect. 2.3).

**Table 1. Overview of input datasets, excluding anthropogenic emissions.**

| Components | | Source | Resolution | |
| | | | Temporal | Spatial |
|---|---|---|---|---|
| Initial and lateral boundary conditions | Meteorological | ERA5 reanalysis | 1h | 0.25°×0.25° |
| | Chemical | CAMS global reanalysis for greenhouse gas | 3h | 0.75°×0.75° |
| Fluxes | Biogenic | Online (VPRM) | Daily | Model resolution |
| | Biomass burning | Fire INventory from NCAR (FINN) v2.5 | Daily | 1km |
| | Ocean | Observation-based global monthly gridded sea surface $pCO_2$ fields | Monthly | 1°×1° |

Biogenic $CO_2$ flux from the vegetation also plays a significant role during the summer. Here, we use the Vegetation Photosynthesis and Respiration Model (VPRM) (Mahadevan et al., 2008), which is coupled online with WRF-GHG. In VPRM, the calculation of net ecosystem exchange (NEE) consists of two components: gross primary production (GPP) and respiration (Res). Since vegetation photosynthesis acts as a sink for $CO_2$, GPP is represented as a negative flux in the WRF-GHG model calculations.

$$NEE = GPP + R_{res} \,, \tag{2}$$



$$GPP = -\lambda T_{scale} \cdot W_{scale} \cdot P_{scale} \cdot EVI \cdot \frac{PAR}{1+\frac{PAR}{PAR_0}} , \qquad (3)$$

$$R_{res} = \alpha T_s + \beta . \qquad (4)$$

Here, Photosynthetically Active Radiation (PAR) is assumed to be approximately equal to the downward shortwave radiation (SW) in the WRF-GHG model (i.e., $PAR \approx SW$). The 2m temperature ($T_s$) and PAR are provided by WRF simulations. $T_{scale}$ represents the temperature sensitivity of photosynthesis, which is defined by a minimum, maximum and optimum temperature ($T_{min}$, $T_{max}$, $T_{opt}$) for photosynthesis for each vegetation class (Evergreen Forest, Deciduous Forest, Mixed Forest, Shrubland, Wetland, Cropland, Grassland and Other.). $\lambda$, $PAR_0$, $\alpha$, $\beta$ are four parameters that depend on the vegetation class.

Considering their importance, and following sensitivity tests (not shown here), these parameters (see Table A1) are adopted from Table 3 in Glauch et al. (2025) taking into account that they use $PAR \approx 0.505 \times SW$, which differs from the default setting $PAR \approx SW$ in WRF-GHG used here. $W_{scale}$ and $P_{scale}$ represent the effect of water stress and leaf age on photosynthesis, respectively. They are both calculated using the Land Surface Water Index (LSWI) (Xiao et al., 2004). Here, the Enhanced Vegetation Index (EVI) and LSWI are derived from the surface reflectance values of 500-m-resolution Moderate

Resolution Imaging Spectroradiometer (MODIS) (Huete et al., 2002; Gao., 1996). The Copernicus Dynamic Land Cover Collection 3 (Buchhorn et al., 2020) with a high spatial resolution of 100 meters is used to calculate the fraction of each vegetation class in every continental grid cell. The VPRM Preprocessor class in pyVPRM was used to generate the input data needed in VPRM (Glauch, et al., 2025).

**2.3 Anthropogenic emission settings**

Considering the importance of anthropogenic emissions inventories and the availability of multiple options, we conducted a sensitivity analysis of anthropogenic $CO_2$ emissions using five different input configurations, based on three different emission inventories, as summarized in Table 2. (1-2) Monthly sector-specific gridmaps of EDGAR v2024, with a spatial resolution of $0.1° \times 0.1°$ (Crippa et al ., 2024).  (3-4) Yearly CAMS-REG-ANT v8.0 sector-specific gridmaps, with a spatial resolution of $0.1° \times 0.05°$ (Kuenen et al., 2022). (5) Yearly TNO_GHGco_v4.1, with a spatial resolution of $1/60° \times 1/120°$ ($\sim 1 \times 1$ km over

central Europe) (Super et al., 2020). As the TNO inventory doesn't encompass the entire simulation domain, emissions in areas outside its coverage are supplemented using the CAMS-REG-ANT inventory. Considering temporal variation, for EDGAR v2024, due to the lack of corresponding sector-specific temporal factors, we assumed constant hourly values within each month. For CAMS-REG-ANT and TNO, we downscaled the yearly fluxes to hourly emissions data, using the sector-specific factors from CAMS-REG-TEMPO (Guevara et al., 2021) and temporal profile factors from Nassar et al. (2013), respectively. In

addition to comparing different emission inventories, we also evaluated the impact of accounting for the height of anthropogenic emission point sources. As pointed out by Brunner et al. (2019), more than 50% of $CO_2$ emissions in Europe are emitted by large point sources, primarily released through stacks and cooling towers, underscoring the importance of accurately representing the vertical distribution of anthropogenic emissions in model simulations. Assuming all anthropogenic emissions are released from point sources, a vertical disaggregation was applied to the sector-specific emission inventories





from (2) EDGAR and (4) CAMS-REG-ANT, using the vertical emission profiles provided by Brunner et al. (2019). The sector mapping between different inventories is detailed in Table A2.

**Table 2. Overview of the five different anthropogenic emissions inputs. The type "S" represents all emissions released at the surface. "P" represents anthropogenic emissions released according to source-specific vertical profiles (Brunner et al., 2019).**

|   | Test | Type | Inventory | Resolution |
|---|------|------|-----------|------------|
| 1 | EDGAR_S | S | EDGAR v2024 (sector-specific) | Monthly 0.1°×0.1° |
| 2 | EDGAR_P | P | | |
| 3 | CAMS_S | S | CAMS-REG-ANT v8.0 (sector-specific) | Yearly 0.1°×0.05° |
| 4 | CAMS_P | P | | |
| 5 | TNO_CAMS | S | TNO_GHGco_v4.1 + CAMS-REG-ANT v8.0 | Yearly 1/60° × 1/120° (for TNO) |

## 3 Observations and methods

We collected observational data for both meteorological and chemical fields, consisting of three types: meteorological observations, in-situ and ground-based remote sensing $CO_2$ observations. Unfortunately, $CO_2$ concentration measurements are lacking within Belgium during this period, therefore, we gathered observational data from the surrounding regions of Belgium.

### 3.1 Synoptic observations in Belgium

A meteorological observation network is operated across Belgium by the Royal Meteorological Institute of Belgium (RMI),
Meteorological Wing of the Air Component of Defense (Meteo Wing), and the Belgian Authority of airways that is the Belgian air navigation and traffic service provider for the civil airspace (Skeyes). To ensure data quality, the synoptic data provided by RMI undergo a quality control (QC) procedure consisting of an automatic process followed by manual supervision, whereas the QC of data from stations belonging to Skeyes and Meteo Wing is conducted independently of RMI (Bertrand et al., 2013). Figure. 2(b) shows the locations of the 21 stations from which observations are currently available, where red triangles
represent 13 stations operated by RMI, blue diamonds represent 7 stations operated by Skeyes, and the orange dot represents 1 station operated by Meteo Wing. Their detailed coordinates are listed in Table A3. These sites are mainly situated on areas covered by short grass and provide hourly near-surface weather parameters, including temperature, wind speed and wind direction. To evaluate the WRF-GHG model performance, the simulated 2-meter temperature (T2) and 10-meter wind speed (WS10) and wind direction (WD10) from the grid nearest to each station within the inner domain were compared with the
observations. It's worth noting that there are eight stations operated by Skeyes and Meteo Wing where wind speed and wind direction observations are recorded only as integer values.



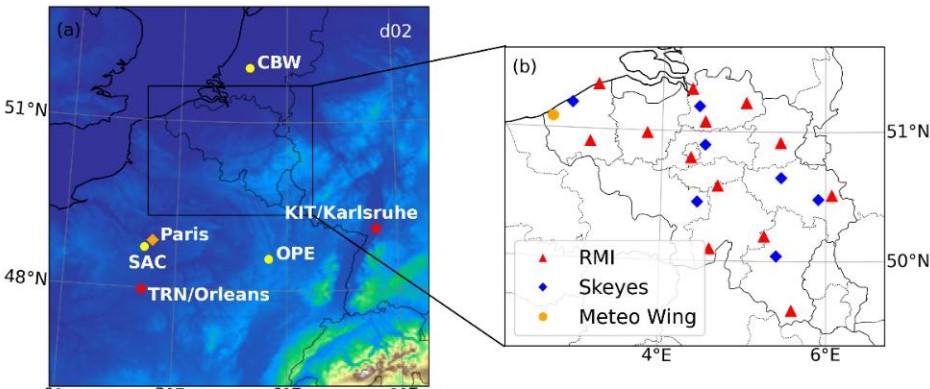

**Figure 2. Locations of ICOS (yellow dots), TCCON (orange diamond) and co-located (red stars) sites within inner domain (a), and synoptic stations in Belgium for which data are available for our study period (b). The background color in (a) represents surface altitude, consistent with Figure 1.**

## 3.2 ICOS – Atmospheric stations

The ICOS (ICOS RI, 2023) atmospheric observation network covers the European region and provides standardized, high-precision scientific data on the carbon cycle and greenhouse gas budgets. It currently comprises 46 stations in 16 countries and there are 5 sites located with available observational data within our inner domain, including Observatoire Pérenne de l'Environnement (OPE), Saclay (SAC), Cabauw (CBW), Trainou (TRN), and Karlsruhe (KIT). Their locations are shown in Fig. 2(a), with details on site coordinates and observation heights listed in Table 3. Each site is equipped with meteorological instruments and Picarro CRDS (Cavity Ring-Down Spectroscopy) GHG analyzers installed at multiple heights on tall towers, providing measurements of hourly meteorological parameters and in situ $CO_2$ mole fractions. For comparison with the simulation results, the simulation data from the inner domain nearest grid cell for each observation site are first extracted. Given that the observation heights at each site do not correspond directly to the model levels, the extracted model data are subsequently interpolated to the corresponding observation heights.

Table 3. The relevant information for each station. 'm a.s.l.' stands for meters above sea level, and 'm a.g.l.' stands for meters above ground level.

| Station | | Location | Altitude (m a.s.l.) | Observation Height (m a.g.l.) |
|---|---|---|---|---|
| Karlsruhe | KIT | 49.0915°N 8.4249°E | 110 | 30/60/100/200 |
| Trainou | TRN | 47.9647°N 2.1125°E | 131 | 50/100/180 |
| Saclay | SAC | 48.7227°N 2.142°E | 16 | 60/100 |
| Observatoire Pérenne de l'Environnement | OPE | 48.5619°N 5.5036°E | 390 | 50/120 |
| Cabauw | CBW | 51.9703°N 4.9264°E | 0 | 207 |





### 3.3 TCCON

TCCON is a global ground-based observation network of Fourier Transform Spectrometers (FTS). It uses the GGG software to retrieve gases mole fractions with high precision and is currently widely used for the validation of satellite measurements (Zhou et al, 2016; Karbasi et al, 2022; Wu et al, 2018). There are three observation sites, Orléans (47.97°N 2.11°E), Karlsruhe (49.10°N 8.44°E), and Paris (48.85°N 2.36°E), located within the inner domain. Among them, the Orléans and Karlsruhe sites are co-located with the ICOS TRN and KIT sites, respectively, and the Paris site is located in an urban area. Each site is

equipped with a Bruker IFS 125HR instrument to record shortwave infrared (SWIR) spectra and use the GGG2020 code to retrieve the column-averaged dry air mole fractions of $CO_2$ ($XCO_2$) (Laughner et al., 2024). TCCON observations are limited to daytime and clear-sky only. To ensure a meaningful comparison with WRF-GHG outputs, the observational data within a 30-minute interval before and after the corresponding model time step are averaged. Additionally, a smoothing correction to account for the a priori profile and averaging kernels (AVKs) associated with the TCCON data was applied to the simulations

data before comparison (Rodgers and Connor, 2003). Details on this correction can be found in Appendix B1 of Callewaert et al. (2022).

### 3.4 Evaluation metrics

To evaluate the performance of the WRF-GHG model, we employed several statistical metrics. The mean bias error (MBE) quantifies the systematic bias between simulations and observations, the standard deviation (STD) of the simulation-

observation differences reflects the variability of the simulations relative to the observations, the root mean square error (RMSE) of the difference quantifies the overall magnitude of simulation uncertainties, and the Pearson correlation coefficient (R) reflects the strength of the expected linear relationship between simulated and observed values. These metrics have been widely used in the assessment of model simulations (e.g. Callewaert et al, 2023; Yarragunta et al, 2025). Their calculation formulas are as follows:

$$Diff_i = mod_i - obs_i ,$$ (5)

$$MBE = \frac{\sum_{i=1}^{N} Diff_i}{N} ,$$ (6)

$$STD = \sqrt{\frac{1}{N}\sum_{i=1}^{N}(Diff_i - MBE)^2} ,$$ (7)

$$RMSE = \sqrt{\frac{1}{N}\sum_{i=1}^{N} Diff_i^2} ,$$ (8)

$$R = \frac{\sum_{i=1}^{N}(mod_i-\overline{mod})(obs_i-\overline{obs})}{\sqrt{\sum_{i=1}^{N}(mod_i-\overline{mod})^2}\cdot\sqrt{\sum_{i=1}^{N}(obs_i-\overline{obs})^2}} ,$$ (9)

where $mod_i$ represents the WRF-GHG simulated values, $obs_i$ represents the observed values, and N is the number of data pairs.



## 4 Model performance

### 4.1 Meteorological fields

The overall evaluation metrics between observed and simulated near-surface temperature, wind speed, and wind direction across all 21 synoptic observation stations in Belgium are given in Table 4, and N represents the number of data pairs. The detailed values at each station can be found in Table A3, and the map is shown in Fig. A1. As mentioned in Sect. 3.1, the wind direction and wind speed observations at eight stations are recorded only as integer values, which limits the ability to capture fine-scale variations and primarily reflect overall trends. Despite the lack of high-precision observations, the WRF model reproduces these variation patterns reasonably well at the trend level. The near-surface temperature was well simulated by the

model, with MBE of 0.06K and R as high as 0.95. Regarding wind fields, the observed wind speed and direction exhibit more pronounced fluctuations than the simulations, as also reported by Zhao et al. (2019). This may result from the rapid changes in real atmospheric conditions, which are difficult for models to capture. Besides, the model tends to slightly overestimate wind speed values in inland areas with dense vegetation cover, which has also been found in previous studies (Duan et al., 2018; Liu et al., 2022; Che et al., 2024). This bias may be attributed to the complex wind distribution in areas with rugged

terrain, where the WRF model fails to adequately account for the additional resistance effects of vegetation on unresolved terrain, ultimately leading to an overestimation of wind speed. In contrast, wind speeds along coastal regions are significantly underestimated, which is probably due to the coastal effects in the WRF simulation (Hahmann et al., 2015).

Table 4. Evaluation metrics between observed and simulated temperature, wind speed, and wind direction across all 21 stations.

|  | N | MBE | STD | RMSE | R |
|---|---|---|---|---|---|
| Temperature (K) | 44426 | 0.06 | 1.61 | 1.61 | 0.95 |
| Wind Speed (m/s) | 43843 | 0.20 | 1.45 | 1.47 | 0.63 |
| Wind Direction (°) | 41983 | -3.31 | 43.16 | 43.29 | 0.57 |

In comparison with ICOS observations, there were no meteorological observations available at KIT and CBW stations, as well as at the 100 m height at TRN site during the simulation period. For the remaining sites, the diurnal cycles of temperature and wind speed, along with the time series of wind direction, are shown in Fig. 3. The values in each subplot represent the corresponding MBE ± STD values between the simulations and the observations. In the diurnal variation plots, the solid lines represent the mean values at the same time of day throughout the simulation period, while the shaded areas indicate the standard

deviation. Similar to the RMI observation sites, the model simulates temperature well at different heights across all ICOS stations, with MBE of less than 0.22 K. As for wind speed, the model captures the diurnal variation at each site, showing higher wind speeds at night and lower speeds during the day. However, it tends to overestimate the wind speed values, especially as the height gets closer to the surface, such as at 50 m at TRN and OPE stations, and at 60 m at SAC station. This agrees with Tuccella et al. (2012), who pointed out that the model can capture the upper-level wind speed profile but tends to overestimate

it in the lower layers. Besides the model tends to significantly overestimate wind speed during the night, as also shown in





previous studies (Zhang and Zheng, 2004; Ngan et al., 2013; Dayal et al., 2020). These wind speed simulation errors may be due to limitations of some parameterization of physical processes during the daytime or nighttime, such as turbulence and surface roughness. Additionally, at the OPE and SAC stations, low-level jets (LLJs) frequently occur near the top of the nocturnal boundary layer (NBL), where excessive downward momentum transport may also lead to overestimated wind speeds

(Zhang and Zheng, 2004). For wind direction, the model can simulate it well, except at the height of 50 m at OPE site. We additionally compared the wind direction observations at the 50 m height at the OPE site, obtained from an independent ICOS instrument through personal communication with the Principal Investigator (PI), with the model simulation results (not shown). Compared to the ICOS data, this independent observation shows better agreement with the model. There might be uncertainties or measurement errors in the wind direction data from the ICOS anemometer at the 50 m height, which could stem from the

sensor itself or be related to its setup, possibly affected by disturbances from the tower structure. The wind speed measurements at the 50m level at OPE are also most probably affected by these limitations.

Overall, the model captures the near-surface variations in meteorological fields well, and exhibits high accuracy in the vertical profiles, which are similar to the performances displayed in previous studies (Tuccella et al., 2012; Mar et al., 2016; Zhao et al., 2023).


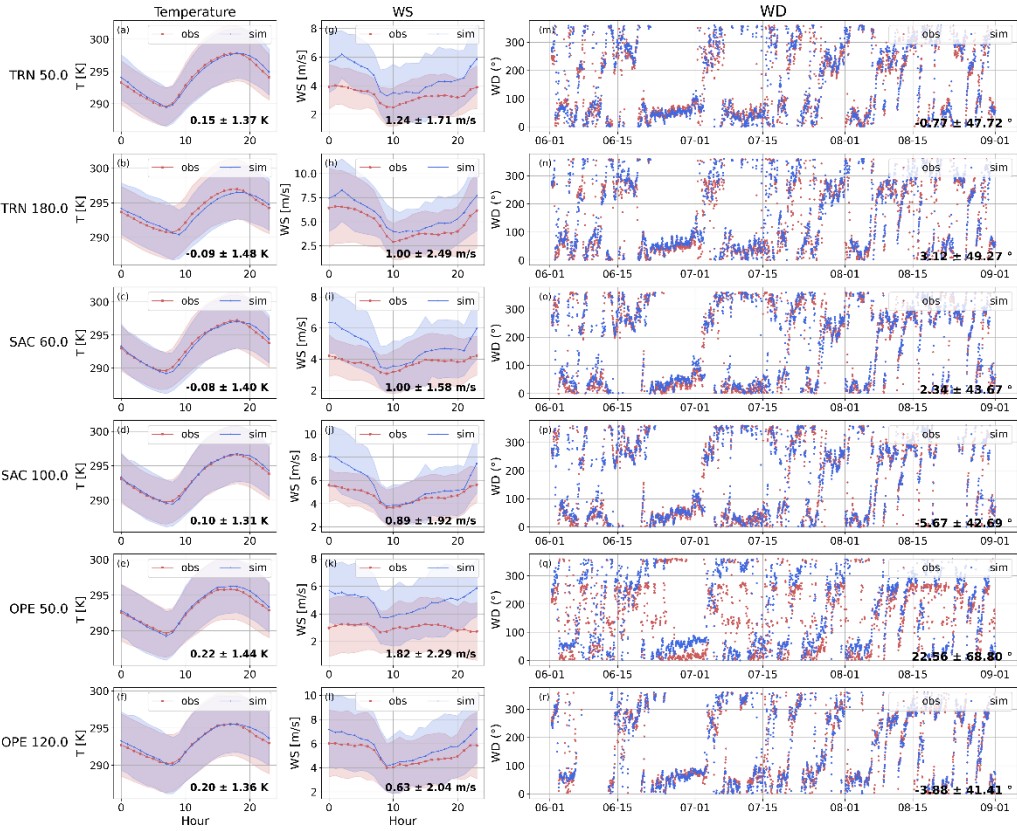

**Figure 3. The diurnal cycles of temperature (a-f) and wind speed (g-l), and the time series of wind direction (m-r) at different heights at various ICOS stations. The time is local time (UTC+2).**



## 4.2 Chemical fields

The simulated $CO_2$ mole fractions are compared with corresponding observations in this section.

### 4.2.1 Comparisons with ICOS – near surface $CO_2$ mole fractions

The statistical metrics for differences of near-surface $CO_2$ mole fractions between observations and simulations at different heights above ground across various ICOS sites using five different anthropogenic emission settings are given in Fig. 4. At most sites, the values of STD and RMSE between the simulations and observations tend to increase at lower heights. Additionally, when anthropogenic emissions are released only at the surface, the differences in simulation results between
emission inventories can be significant. For example, using surface emission from either EDGAR or TNO can lead to differences up to -14.99 ± 31.98 ppm (MBE ± STD) at the KIT site at 30 m height. It reflects that the anthropogenic emission inventory has a significant impact on the simulation of near-surface $CO_2$ mole fractions, especially at lower heights above ground-level and for peri-urban stations.

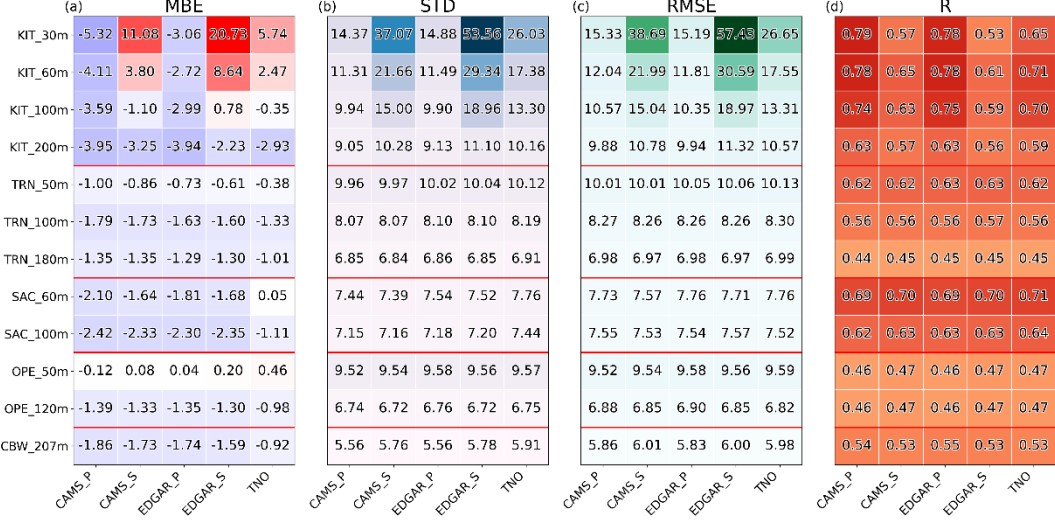

**Figure 4. The MBE (a), STD (b), RMSE (c), and R (d) of near surface $CO_2$ mole fractions between observations and five different simulations at different heights at various ICOS stations. The colors in (a) represent values from negative to positive, with a blue–white–red gradient: negative values in blue, positive values in red, and values near zero in white, whereas in (b–d), more intense colors represent larger values and less intense colors represent smaller values.**

At the KIT site, taking into account the vertical distribution of anthropogenic emissions has a significant impact on the simulation results. Simulations using elevated emissions show much better agreement with observations, especially for the lower levels. At the height of 30 m, when using the CAMS-REG-ANT emission inventory and considering elevated emissions, the model-observation bias has a low MBE of -5.32 ppm, a STD of 14.37 ppm, a RMSE of 15.33 ppm, and high correlation R of 0.79. When only surface emissions are considered, MBE, STD, and RMSE increase significantly to 11.08 ppm, 37.07 ppm,
and 38.69 ppm, respectively, and R drops to 0.57. The simulations using EDGAR v2024 display similar characteristics, with



even larger differences between elevated emissions and surface emissions. As shown in Fig. 5(i-l), compared to simulations accounting for elevated emissions, the simulations that only consider surface emissions tend to overestimate $CO_2$ mole fractions at 30 m, 60 m and 100 m heights. Their diurnal cycles (Fig. 5(a–h)) indicate that all five simulations are able to capture the diurnal variation of $CO_2$ and reproduce the lower $CO_2$ mole fractions well in the afternoon, but the simulation

considering only surface emissions significantly overestimates higher $CO_2$ mole fractions in the morning, leading to large discrepancies with the observations. This overestimation is especially pronounced when using the EDGAR inventory. However, at the other four observation sites (TRN, SAC, OPE, CBW), the differences among the five simulation results are relatively small, and the simulations considering only surface emissions do not show a significant overestimation (see Fig. A2). The discussion in Sect. 5.1 will provide a more detailed analysis of this phenomenon.

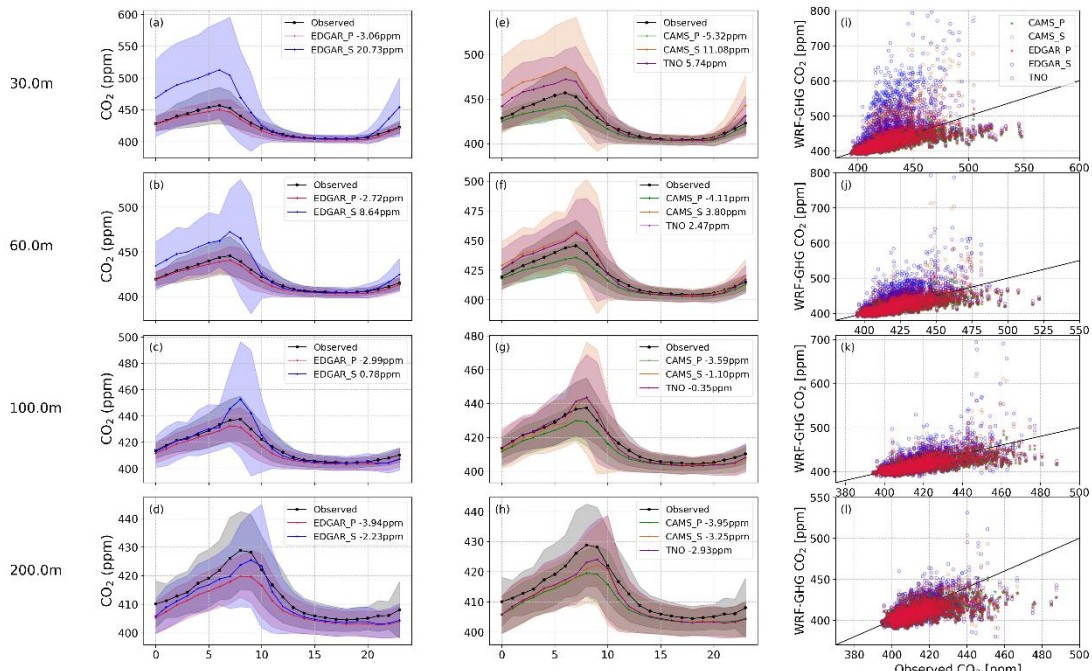


**Figure 5.** Diurnal cycles (local time) of simulations with different anthropogenic emissions and observations (a-h), where the values represent the MBE between the observations and each simulation, along with scatterplots comparing each simulation to the observations (i-l) at different heights at the ICOS KIT site.

As shown in Fig. 6, the contributions from biomass burning and oceanic sources to the diurnal cycles are negligible at these five sites. At the KIT, SAC and CBW sites, the diurnal variability of near-surface $CO_2$ mole fractions is mainly driven by local anthropogenic emissions and biogenic processes, with the biogenic signal systematically stronger. The diurnal variation of anthropogenic $CO_2$ is attributed to planetary boundary layer (PBL) dynamics and regional transport. Between 15:00 and 17:00 in the afternoon, the PBL (see Fig. A3) reaches its maximum height, coinciding with the minimum anthropogenic $CO_2$ mole

fractions. After sunset, the PBL gradually decreases and remains shallow from midnight until sunrise, leading to the accumulation of anthropogenic emissions. The mole fractions of $CO_2$ increase during this period and typically peak between





06:00 and 10:00 in the morning. As the PBL rises again after sunrise and turbulent mixing emerges, $CO_2$ mole fractions begin to decrease, forming a distinct diurnal cycle. The diurnal variation of biogenic processes is driven by vegetation photosynthesis during the day and respiration at night. Biogenic processes are the dominant drivers of the $CO_2$ mole fractions at the OPE and

TRN sites.  These results align with Storm et al. (2023), which found that in the summer of 2020, the biogenic flux signals at most ICOS sites were stronger than anthropogenic emissions, and the largest signal is associated with cropland.

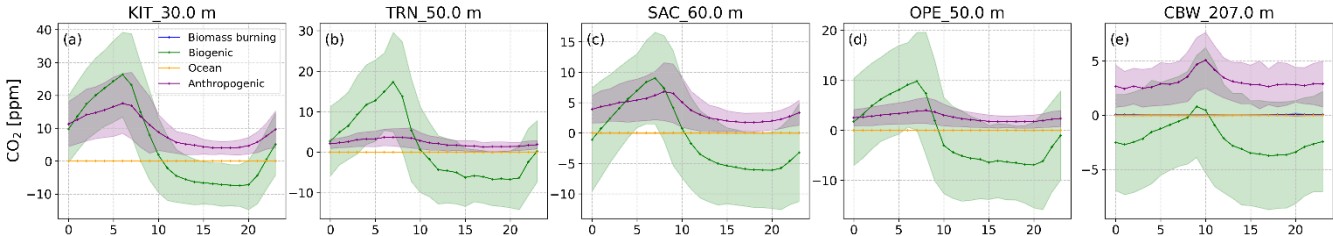

**Figure 6. Diurnal cycles (local time) of simulated tracer contributions at 30 m at the KIT (a), 50 m at the TRN (b), 60 m at the SAC (c), 50 m at the OPE (d) and 207 m at the CBW (e) sites. Here, the anthropogenic emissions are based on EDGAR v2024, taking**
**into account the vertical emission profiles.**

### 4.2.2 Comparison with TCCON – XCO₂

Figure 7b shows the time series of observed and simulated $XCO_2$ at three TCCON sites, Figure 7c is the corresponding scatterplot, with N representing the number of data pairs, and Figure 7a shows the differences between observations and simulations at each site. Here the simulations are based on the TNO inventory. Due to the lack of observational data at the

Paris and Karlsruhe sites in June, the number of valid data pairs for comparison with the simulations is only 123 and 243, respectively. It is obvious that at the Orleans site, the simulated $XCO_2$ values show a significant underestimation in early June, which will be discussed in detail in the following section.

The statistical metrics between the observed and simulated $XCO_2$ using five different anthropogenic emission inputs at the three TCCON sites are given in Fig. 8. Among the three sites, the choice of emission inventory has the largest impact on the

simulations at the Paris site. At this site, the difference in simulated $XCO_2$ between using the EDGAR and TNO inventories reaches $0.50 \pm 0.34$ ppm. Among the five sensitivity tests, all the simulations show an overestimation, with MBEs around 1.2 ppm, and up to 1.65 ppm when using TNO. An inspection of the different tracer contributions to $XCO_2$ (see Fig. 9) indicates that despite uncertainties in the estimation of biogenic carbon fluxes, the overestimation of background $XCO_2$ from the CAMS and anthropogenic emissions plays a dominant role in causing biases in the simulated $XCO_2$. During the period from late July

to early August, the simulations show an overestimation, consistent with the relatively large contribution from anthropogenic tracers during that period. These results indicate that the uncertainty in anthropogenic emission inventories remains relatively high in urban areas (Gately and Hutyra, 2017; Super et al., 2020).



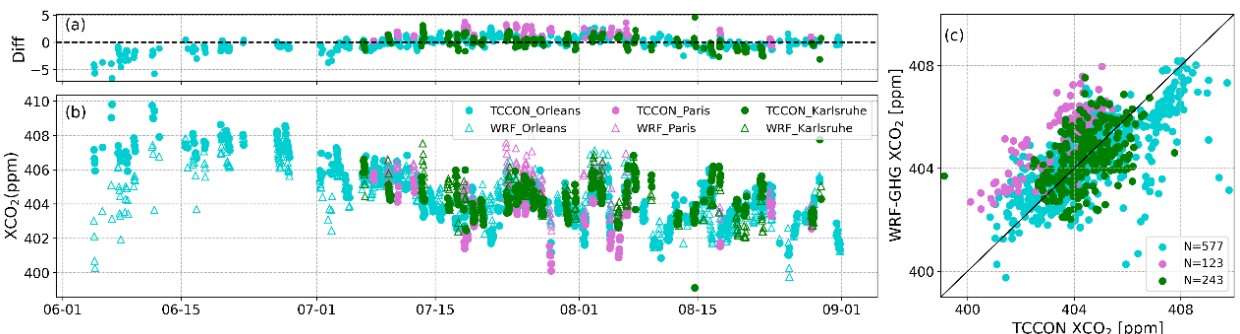

**Figure 7. Time series (local time) of observed and simulated $XCO_2$ at three TCCON sites using TNO inventory (b), their absolute differences (WRF-GHG - TCCON) (a), and scatterplots (c).**

Additionally, as $XCO_2$ is less sensitive to vertical transport processes (Wunch et al., 2011), one expects less impact of considering elevated anthropogenic emissions heights on the simulation results for $XCO_2$. Indeed, at the Orléans and Paris sites, the impact is negligible for a given emission inventory, while at the Karlsruhe site, we observe a small improvement in the simulation of $XCO_2$ but not as notable as that observed for near-surface mole fractions. At the Karlsruhe site, when accounting for elevated emission heights in the CAMS-REG-ANT inventory, the STD and RMSE of differences between simulations and observations improves from 1.13 ppm to 1.05 ppm and 1.06 ppm, respectively while R increases from 0.44 to 0.47; for EDGAR, STD and RMSE decrease from 1.16 ppm to 1.03 ppm and 1.04 ppm, respectively, while R increases from 0.43 to 0.48.

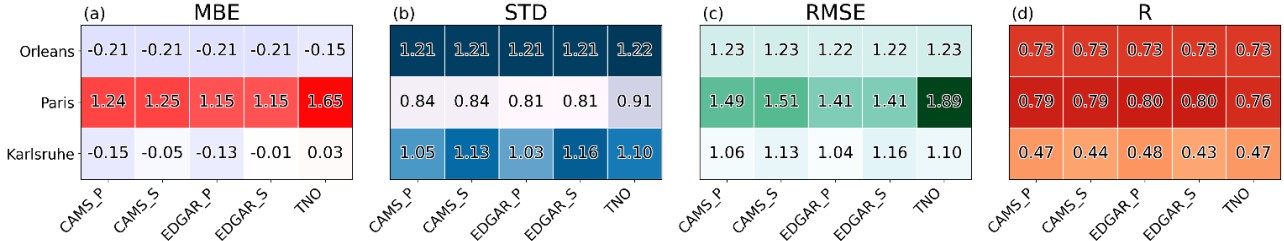

**Figure 8. The MBE (a), STD (b), RMSE (c), and R (d) of $XCO_2$ between observations and five different simulations at three TCCON stations.**



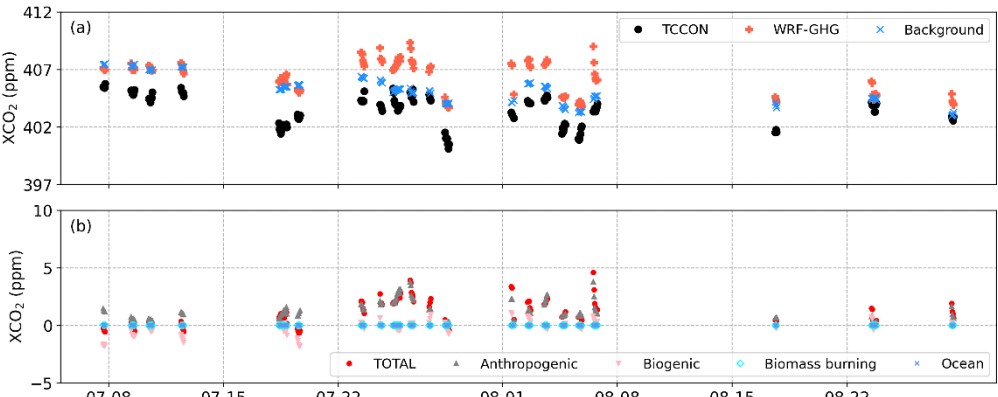

**Figure 9. Time series (local time) of (a) observed and simulated total, background, and (b) tracer-specific XCO₂ at Paris site. The simulated values here are all without AVK smoothing.**

## 5 Discussion

We will first discuss the anthropogenic emissions sensitivity tests results and then focus on the biogenic fluxes impacts on model biases reported in June 2018.

### 5.1 Impact of taking into account the height of anthropogenic emissions

As previously noted, at the KIT site, whether anthropogenic emissions are released according to source-specific vertical profiles or only at the surface has a significant impact on the simulation of near-surface $CO_2$ mole fractions, and also shows a slight influence on $XCO_2$. Such large impacts are not observed at the other sites.

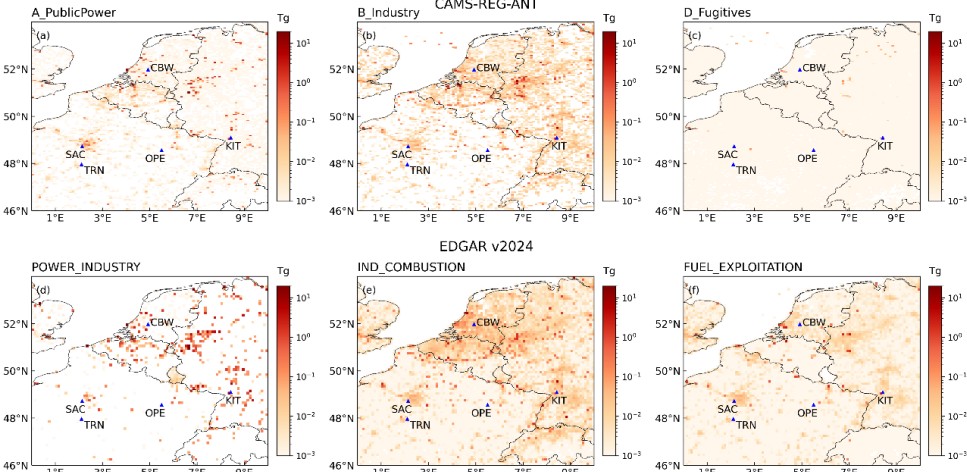

**Figure 10. The maps of sector-specific CO2 emissions for 2018 from CAMS-REG-ANT (a) public power, (b) industry and (c) fugitives sectors, and EDGAR v2024 (d) power industry, (e) industrial combustion and (f) fuel exploitation. All values are shown on a logarithmic scale (base 10, unit: Tg).**





Figure 10 shows the 2018 emission maps of the major contributing sectors from two inventories, including the Public Power (SNAP 1), Industry (SNAP 3) and Fugitives (SNAP 5) sectors from CAMS-REG-ANT, and the Power Industry (SNAP 1), Industrial Combustion (SNAP 3) and Fuel Exploitation (SNAP 5) sectors from EDGAR. Although monthly EDGAR emission
data for 2018 were used in the simulations, in these maps the monthly EDGAR inventories were aggregated by summing over all months to produce an annual emission inventory for 2018 for enabling comparison with the CAMS-REG-ANT inventory. It is clearly evident that, except for Industrial Combustion, there are large emission sources near the KIT site, whereas no obvious emissions are observed near the other sites. According to Google Maps, we found that approximately 6.5 km southwest in a straight line from the KIT observation site lies the largest oil refinery of Germany (Junkermann et al., 2011), and about 45
km north of the site there is a gas-fired combined heat and power (CHP) plant located within the Badische Anilinund Sodafabrik (BASF) chemical production facility in Ludwigshafen, Germany. These two are likely the main contributors to the emissions in the inventory near KIT site.

In addition, based on footprint simulations from the regional Stochastic Time-Inverted Lagrangian Transport (STILT) model (Lin et al., 2003), we calculated the aggregated footprints for each afternoon hour (14:00 LT) over the simulation period (not
shown here). The results show that, centered on the KIT site, at least 80% of $CO_2$ enhancements from anthropogenic emissions are covered within a radius of approximately 1.5°. Therefore, we calculated the $CO_2$ emission fluxes from the sectors corresponding to those in Fig. 10 in 2018, within the area surrounding the KIT site, bounded by 47.6°N to 50.6°N and 6.94°E to 9.94°E (see Fig. 11a). For consistency, the SNAP sector classification is used here. The vertical emission profiles applied for the corresponding sectors are given in Fig. 11b, and the complete vertical profiles for all sectors can be found in Fig. 2a of
Brunner et al. (2019). For the EDGAR inventory, emissions from sector SNAP 1 are the dominant source, with the emission flux in 2018 as high as 52.18 Tg, and for the CAMS-REG-ANT inventory, emissions from sectors SNAP 1 and SNAP 3 are the main sources, with fluxes of 24.39 Tg and 32.35 Tg, respectively. According to their respective vertical emission profiles, emissions from SNAP 1 and SNAP 3 are primarily concentrated at higher altitudes. In particular, the emission profile of SNAP 1 begins at approximately 150 meters above ground level, without emissions near the surface. This can explain the finding that
in Sect. 4.2.1 at the KIT 30 m height, the difference between tests S and P is larger for EDGAR v2024 than for CAMS-REG-ANT.

In summary, at KIT, where anthropogenic emissions are significant, SNAP 1 and SNAP 3 are the dominant emissions sectors explaining the improvement of the simulation results by including vertical emissions profiles. Considering anthropogenic vertical profile emissions in the model setup significantly improve the simulations performance, especially in the vicinity of
large sources.




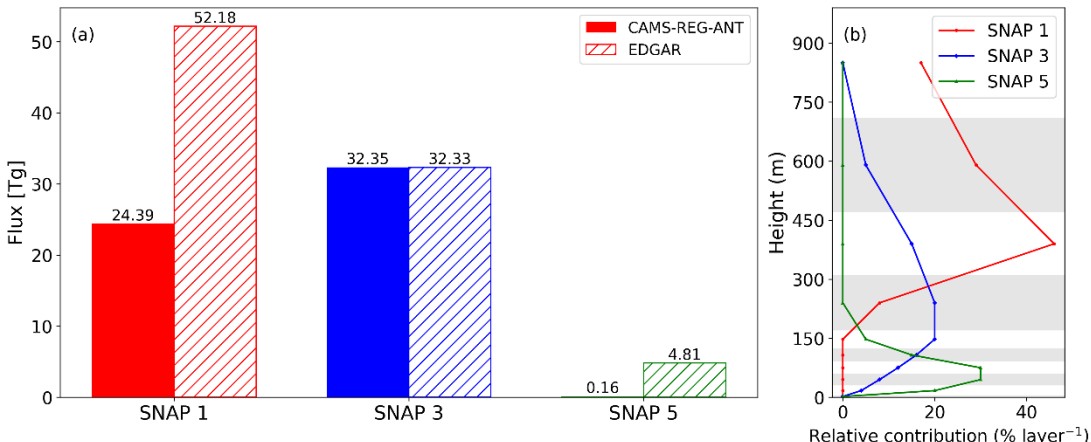

**Figure 11. (a) CO₂ emission fluxes from different sectors in 2018 within the area (47.6°N - 50.6°N, 6.94°E - 9.94°E) surrounding the KIT site, and (b) the vertical emission profiles applied for the corresponding sectors, the height refers to the altitude above ground level.**

## 5.2 Underestimation of model CO₂ simulations in early June

A significant underestimation of the simulated XCO₂ was found at the Orleans site in early June. According to the contributions of each tracer to the total simulated XCO₂ (see Fig. 12), a similar pattern is found in the biogenic component. Additionally, a comparison between observed and simulated near-surface CO₂ mole fractions at various heights from the co-located ICOS TRN site also reveals a clear underestimation in early June (Fig. A4). This site pair is hereafter referred to as Orleans/TRN.

Figure 12c shows the vertical distribution of the biogenic CO₂ tracer at Orleans/TRN over time, revealing a sink spanning a large vertical extent in early June. At the other two TCCON sites (Paris and Karlsruhe), due to the lack of observational data in June, it is difficult to determine whether a similar feature exists. However, the underestimation of CO₂ mole fractions in early June is not confined to the Orleans/TRN site. A consistent underestimation during this period is also observed at the other ICOS sites (OPE, SAC, CBW, see Fig. A4), except for the KIT site which including significant anthropogenic emissions, indicating that this is likely a regional feature rather than a localized effect at a single site. In addition, the STILT simulation results published on the ICOS Carbon Portal also exhibit an underestimation at these ICOS sites (except for the KIT site) in early June. The biospheric fluxes are from the VPRM model (Gerbig and Koch, 2023), with VPRM parameters optimized for the year 2007 using 46 sites across Europe, and the land cover classification is based on SYNMAP.



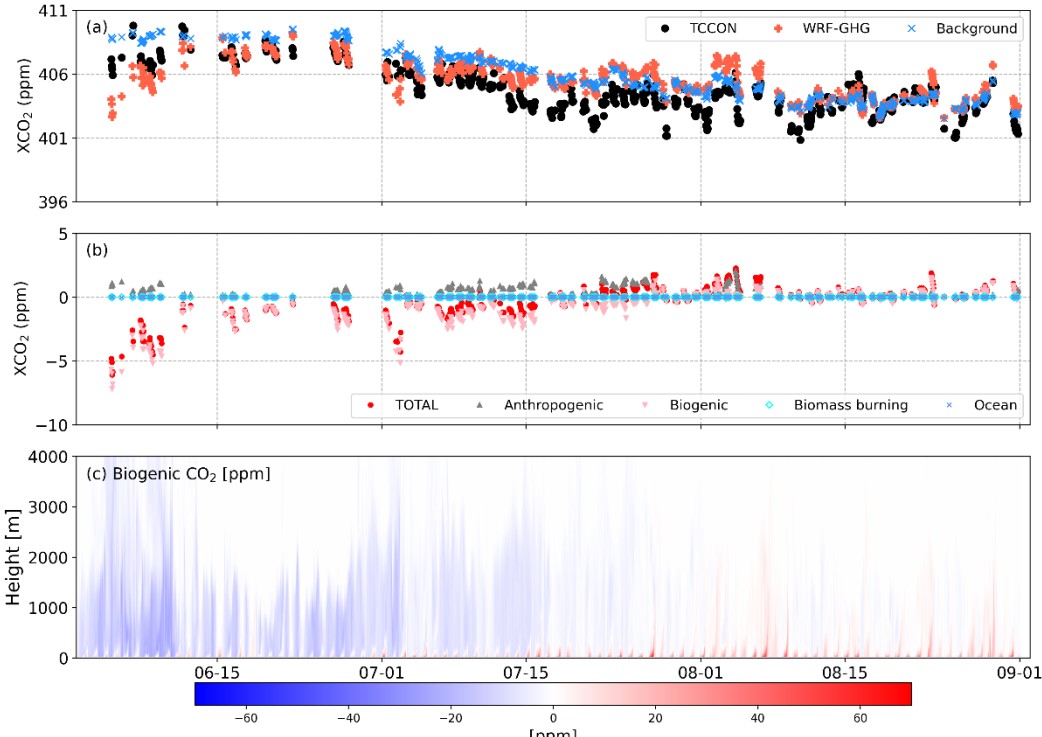

**Figure 12. Time series (local time) of (a) observed and simulated total, background, (b) tracer-specific XCO₂, and (c) variation of simulated biogenic CO₂ mole fractions over time (local time) and height above ground at Orleans/TRN site. The simulated values here are all without AVK smoothing.**

At most ICOS and TCCON sites included in this study, GPP and daytime NEE in June exhibit a stronger carbon sink compared to July and August. Starting from July, LSWI and EVI values, which are used as input to VPRM, across all vegetation types declined significantly, reflecting an overall decrease in vegetation activity and greenness. This trend was observed at the Orleans/TRN site (see Fig. A5), and similar patterns were found at other sites. This could be attributed to a prolonged and intense drought that occurred in Europe during the summer of 2018. This event directly affected temperature, soil moisture, precipitation, and ecosystem functioning (Buras et al., 2020). Simulations by WRF-GHG and VPRM, driven by meteorological fields and vegetation indices, both responded to this drought event.

Regarding the mismatch between the model and observations in early June, the simulated NEE shows a pronounced carbon sink at most sites. Unfortunately, the lack of co-located flux measurements at these atmospheric observation sites limits the direct evaluation of flux accuracy. Alternatively, we evaluated the four ICOS flux stations within the model's inner domain: two sites in Germany, DE_RuR (50.621914N, 6.304126E; grasslands) and DE_RuW (50.50493N, 6.330963E; forest), and two in France, FR_EM2 (49.87211N, 3.02065E; crops) and FR_LGt (47.322918N, 2.284102E; acid fen) (not shown here). The simulated RES shows a significant underestimation at all sites, except at FR_LGt site, which is consistent with Hu et al. (2021), who pointed out that the VPRM model tends to underestimate ecosystem respiration during the growing season, which could introduce biases in the simulation of NEE. At the two French sites, the simulated GPP exhibits more gross uptake significantly.



Additionally, we tested four different sets of VPRM parameter settings (not show here), all of which resulted in a similar underestimation in early June, preliminarily ruling out parameter settings as the primary source of the bias. It could rather be

the consequences of the limitations either of the model itself.

In summary, the underestimation of simulated $CO_2$ mole fractions observed at multiple ICOS and TCCON sites in early June might be attributed to an underestimation of ecosystem respiration fluxes or an overestimation of GPP. However, owing to the lack of co-located flux measurements, the exact source of this bias remains unclear and requires further investigation.

## 6 Conclusion

In this study, we simulated the spatiotemporal distribution of $CO_2$ mole fractions over Belgium and Western Europe during the summer (June to August) of 2018 using WRF-GHG. Given the significance of anthropogenic emissions and the diversity of emission inventories, we conducted several sensitivity tests and evaluated the model performance from both meteorological and chemical field perspectives by comparing with ground-based observations at multiple ICOS and TCCON stations, on the purpose of optimizing the modeling set-up and – at a later stage - analyze the $CO_2$ concentration variations over western Europe.

Overall, the WRF-GHG model reproduces the meteorological fields well, especially temperature, with high Pearson correlation coefficients ranging from 0.92 to 0.96 against observations from various synoptic sites. The model can capture the variations of wind speed at different heights, on both daily and monthly timescales. Moreover, the diurnal variation of near-surface $CO_2$ mole fractions at different heights across the five ICOS observation sites was well captured by the model. During the summer 2018, variations in $CO_2$ mole fractions across Western Europe were mainly influenced by anthropogenic emissions and

biogenic fluxes.

Sensitivity tests indicate that near large anthropogenic emission sources, the simulated near-surface $CO_2$ mole fractions are highly sensitive to the choice of anthropogenic emission inventory and the adoption of vertical emission profiles. At the KIT site in Germany, which is located near a very large oil refinery and power plant, differences between emission inventories can lead to discrepancies of up to $-14.99\pm31.98$ ppm in simulated near surface $CO_2$ mole fractions. In addition, releasing

anthropogenic emissions based on source-specific vertical profiles can significantly improve the accuracy of simulations, which is likely due to a more realistic representation of the real emissions. In contrast, at other observation sites where surrounding anthropogenic emissions are relatively low, the impact of vertical emission profiles on simulation results is much smaller.

Regarding $XCO_2$, the model seems to be less sensitive to the choice of emission inventories and anthropogenic emission

heights, but certain effects are still evident. At the Paris urban site, all simulations overestimate $XCO_2$ by approximately 1.2– 1.6 ppm. This bias is primarily attributed to the uncertainties introduced by anthropogenic emissions and using CAMS data as initial and boundary conditions. In addition, the differences caused by different anthropogenic emission inventories can reach up to $0.50 \pm 0.34$ ppm. Although it is less pronounced than for the near-surface mole fractions at the ICOS site, the simulation



results confirm that considering vertical emission profiles leads to a modest improvement in model simulations at the Karlsruhe
site.

Additionally, biogenic fluxes contribute significantly to $CO_2$ mole fractions during the growing season. The large negative
bias observed in WRF-GHG simulations of $CO_2$ mole fractions in early June at most ICOS and TCCON sites may be attributed
to the underestimation of RES or the overestimation of GPP by the VPRM model. However, due to the lack of flux observations,
the exact cause remains uncertain.

This study demonstrates the feasibility of using the WRF-GHG model to simulate $CO_2$ concentration variations over Western
Europe. However, to further improve simulation accuracy, future efforts should focus on optimizing boundary conditions and
refining the construction of source-specific vertical emission profiles. Additionally, due to the relatively simple
parameterization of the current VPRM, it may not perform well under all conditions, whether under normal climate or extreme
stress. For example, biases in respiration fluxes may arise from nonlinear ecosystem responses to extreme temperature and
moisture conditions, which are not accounted for in the model. Therefore, a modified VPRM model is necessary (Gourdji et
al., 2022).  At the same time, observational data play a critical role in model evaluation, the availability of time-synchronized
ground-based observations of $CO_2$ fluxes and concentrations can help assess the performance of the model, thereby enhancing
the credibility and scientific interpretation of the simulation results.





# Appendix

Table A1. Overview of the VPRM parameters. The temperature parameters are all in degrees Celsius. The abbreviations are as follows: EF - evergreen forest, DF - deciduous forest, MF - mixed forest, SHR - shrubland, WET - wetland, CRO - cropland, GRA - grassland.

|  |  | Tmin | Topt | Tmax | Tlow | λ | Par0 | α | β |
|---|---|---|---|---|---|---|---|---|---|
| 1 | EF | -4 | 15 | 38 | -3 | -0.26 | 263.60 | 0.21 | 1.15 |
| 2 | DF | 1 | 21 | 37 | 0 | -0.26 | 252.90 | 0.23 | 1.26 |
| 3 | MF | -1 | 18 | 38 | 0 | -0.28 | 227.81 | 0.19 | 0.93 |
| 4 | SHR | -1 | 19 | 44 | 2 | -0.20 | 224.27 | 0.08 | 0.56 |
| 5 | WET | -2 | 26 | 40 | 0 | -0.24 | 201.85 | 0.3 | -0.39 |
| 6 | CRO | -3 | 16 | 50 | -3 | -0.18 | 485.20 | 0.17 | 1.14 |
| 7 | GRA | -2 | 17 | 36 | -2 | -0.44 | 223.92 | 0.27 | 1.63 |
| 8 | OTHER | 0 | 0 | 0 | 0 | 0 | 0 | 0 | 0 |

Table A2. Sector mapping between different emission inventories (Granier et al., 2019)

| SNAP | CAMS-REG-ANT (GNFR) | EDGAR v2024 |
|---|---|---|
| 1 Energy industry | A Public Power | Power Industry |
| 2 Non-industrial combustion | C Other Stationary Comb | Buildings |
| 3 Combustion in manufacturing industry | B Industry | Industrial combustion |
| 4 Production processes |  | Processes |
| 5 Extraction of fossil fuels | D Fugitive | Fuel exploitation |
| 6 Product use | E Solvents |  |
| 7 Road transport | F Road Transport |  |
| 8 Non-road transport | G Shipping<br>H Aviation<br>I Offroad | Transport |
| 9 Waste treatment | J Waste | Waste |
| 10 Agriculture | K Agrilivestock<br>L AgriOther | Agriculture |


Table A3. Evaluation metrics for temperature, wind speed, and wind direction between observations and simulations at the 21 synoptic stations. The eight stations marked with * are operated by Skeyes and Meteo Wing, where wind speed and wind direction observations are recorded only as integer values. The three stations marked with [c] represent coastal sites.

| Site | Location | Temperature | | | | | Wind Speed | | | | | Wind Direction | | | | |
|---|---|---|---|---|---|---|---|---|---|---|---|---|---|---|---|---|
|  |  | N | MBE | STD | RMSE | R | N | MBE | STD | RMSE | R | N | MBE | STD | RMSE | R |
| 6400*[c] | 51.088N | 2200 | -0.24 | 1.27 | 1.29 | 0.95 | 2201 | -0.32 | 1.36 | 1.40 | 0.75 | 2201 | -1.14 | 51.56 | 51.57 | 0.61 |





| ID | Coord | | | | | | | | | | | | | | | |
|---|---|---|---|---|---|---|---|---|---|---|---|---|---|---|---|---|
| | 2.652E | | | | | | | | | | | | | | | |
| 6407*c | 51.200N 2.887E | 2202 | -0.11 | 1.84 | 1.84 | 0.92 | 2202 | -1.19 | 1.65 | 2.04 | 0.67 | 2202 | 3.40 | 37.29 | 37.45 | 0.67 |
| 6414 | 50.904N 3.122E | 2180 | -0.63 | 1.56 | 1.68 | 0.95 | 2180 | 0.21 | 1.13 | 1.15 | 0.75 | 2149 | 5.38 | 36.05 | 36.46 | 0.58 |
| 6418c | 51.347N 3.202E | 2135 | -0.38 | 1.31 | 1.36 | 0.93 | 2134 | -2.42 | 1.55 | 2.88 | 0.74 | 2134 | 4.99 | 32.29 | 32.68 | 0.68 |
| 6434 | 50.980N 3.816E | 2202 | -0.50 | 1.63 | 1.71 | 0.95 | 2202 | 0.40 | 1.16 | 1.22 | 0.72 | 2123 | 0.47 | 41.14 | 41.15 | 0.58 |
| 6438 | 51.325N 4.364E | 2189 | -0.14 | 1.47 | 1.48 | 0.95 | 2189 | -0.56 | 1.27 | 1.39 | 0.70 | 2100 | 7.38 | 39.57 | 40.25 | 0.61 |
| 6439 | 51.075N 4.525E | 1624 | 0.19 | 1.40 | 1.41 | 0.96 | 1621 | -0.20 | 0.91 | 0.93 | 0.77 | 1523 | 4.19 | 41.10 | 41.32 | 0.50 |
| 6447 | 50.797N 4.358E | 2202 | 0.06 | 1.42 | 1.42 | 0.95 | 2202 | -0.37 | 0.96 | 1.03 | 0.69 | 2193 | 4.90 | 40.50 | 40.81 | 0.62 |
| 6449* | 50.454N 4.442E | 2202 | 0.30 | 1.53 | 1.56 | 0.95 | 2202 | -0.98 | 1.14 | 1.50 | 0.69 | 2096 | 4.00 | 37.09 | 37.31 | 0.56 |
| 6450* | 51.191N 4.452E | 2202 | 0.01 | 1.41 | 1.41 | 0.95 | 2202 | -1.07 | 1.20 | 1.61 | 0.69 | 2072 | -5.15 | 38.94 | 39.29 | 0.62 |
| 6451* | 50.896N 4.527E | 2202 | 0.99 | 1.46 | 1.76 | 0.95 | 2202 | -0.82 | 1.21 | 1.46 | 0.68 | 2064 | -4.61 | 38.67 | 38.94 | 0.60 |
| 6455 | 50.095N 4.595E | 2202 | -0.29 | 1.86 | 1.89 | 0.94 | 2202 | 0.72 | 1.08 | 1.30 | 0.61 | 1984 | 10.96 | 53.46 | 54.52 | 0.48 |
| 6459 | 50.582N 4.689E | 2202 | -0.06 | 1.48 | 1.49 | 0.96 | 2202 | 0.15 | 1.17 | 1.18 | 0.71 | 2183 | 15.95 | 40.30 | 43.34 | 0.49 |
| 6464 | 51.221N 5.027E | 2202 | -0.45 | 1.58 | 1.65 | 0.96 | 2202 | 0.98 | 1.12 | 1.49 | 0.59 | 1918 | 8.15 | 41.60 | 42.40 | 0.50 |
| 6472 | 50.194N 5.255E | 2202 | 0.23 | 1.72 | 1.73 | 0.94 | 2202 | 0.23 | 1.10 | 1.12 | 0.74 | 2149 | 9.15 | 49.61 | 50.46 | 0.50 |
| 6476* | 50.039N 5.404E | 2202 | -0.19 | 1.49 | 1.50 | 0.95 | 1627 | 0.09 | 1.15 | 1.16 | 0.71 | 1497 | -1.32 | 40.81 | 40.84 | 0.65 |
| 6477 | 50.915N 5.450E | 2202 | -0.38 | 1.89 | 1.93 | 0.94 | 2202 | 0.43 | 1.18 | 1.26 | 0.69 | 2029 | 0.86 | 47.55 | 47.55 | 0.54 |
| 6478* | 50.645N | 2202 | 0.43 | 1.46 | 1.52 | 0.95 | 2200 | -0.72 | 1.18 | 1.38 | 0.71 | 2084 | 4.98 | 43.18 | 43.48 | 0.58 |



| | | | | | | | | | | | | | | | | | |
|---|---|---|---|---|---|---|---|---|---|---|---|---|---|---|---|---|---|
| | 5.459E | | | | | | | | | | | | | | | | |
| 6484 | 49.620N 5.587E | 1147 | 0.42 | 1.98 | 2.02 | 0.94 | 1147 | 0.44 | 1.26 | 1.34 | 0.47 | 1137 | -19.34 | 56.21 | 59.47 | 0.48 |
| 6490* | 50.479N 5.910E | 2183 | -0.15 | 1.54 | 1.55 | 0.95 | 2174 | 0.35 | 1.24 | 1.29 | 0.65 | 2019 | -4.65 | 46.05 | 46.29 | 0.47 |
| 6494 | 50.511N 6.073E | 2121 | -0.19 | 1.49 | 1.51 | 0.95 | 2126 | 0.78 | 1.03 | 1.30 | 0.71 | 2105 | 9.10 | 41.32 | 42.32 | 0.55 |

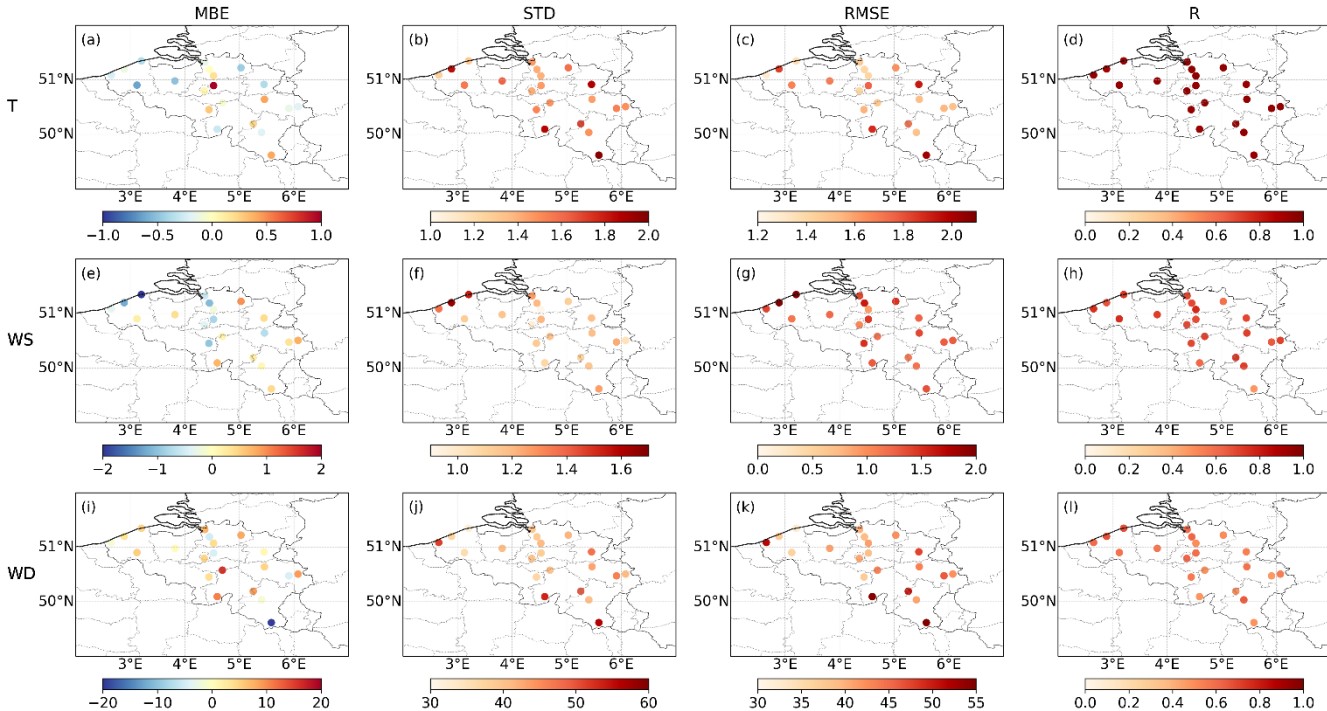


**Figure A1. Map of evaluation metrics for temperature (a-d), wind speed (e-h), and wind direction (i-l) between observed and simulated values at 21 synoptic stations.**



**Figure A2. Diurnal cycles (local time) of simulations with different anthropogenic emissions and observations at 50 m at the TRN (a, b), 60 m at the SAC (c, d), 50 m at the OPE (e, f) and 207 m at the CBW (g, h) sites, where the values represent the MBE between the observations and each simulation.**






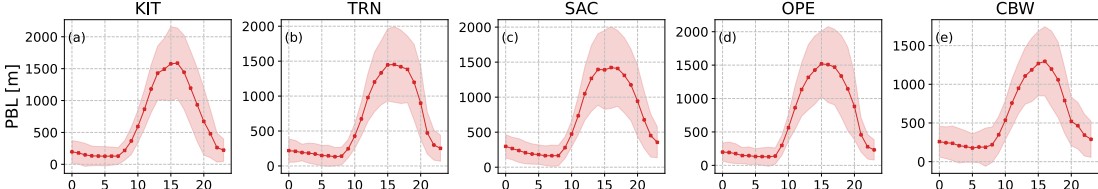

**Figure A3. The diurnal cycles (local time) of planetary boundary layer height simulated by WRF-GHG at the KIT (a), the TRN (b), the SAC (c), the OPE (d) and the CBW (e) sites.**

**KIT 30.0 m**

5.74±26.03 ppm

**TRN 50.0 m**

-0.38±10.12 ppm

**SAC 60.0 m**

0.05±7.76 ppm

**OPE 50.0 m**

0.46±9.57 ppm

**CBW 207.0 m**

-0.92±5.91 ppm


**Figure A4. Time series of the between observed and the simulated near surface CO2 mole fractions using TNO inventory at 30 m at the KIT (a), 50 m at the TRN (b), 60 m at the SAC (c), 50 m at the OPE (d) and 207 m at the CBW (e) sites, where the values represent the MBE ± STD between the observations and simulations. The red curve represents the 24-hour moving average.**



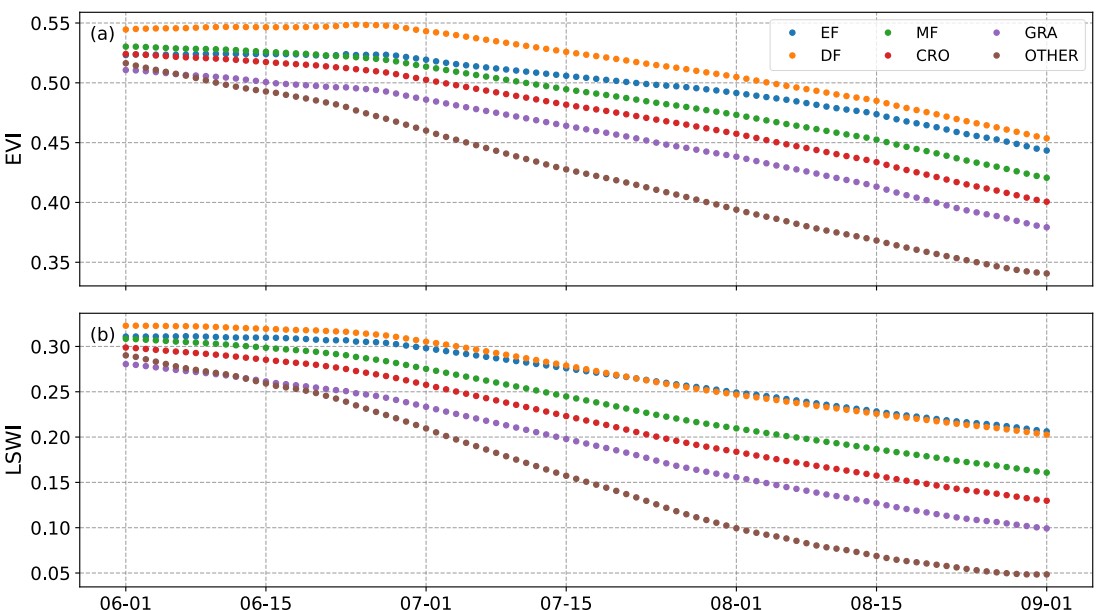

**Figure A5. Time series of EVI (a) and LSWI (b) for different vegetation types from MODIS at Orleans/TRN site. According to the land cover data provided by Copernicus, there are no Shrubland (SHR) and Wetlands (WET) vegetation types near Orleans/TRN. Therefore, their corresponding EVI and LSWI values are 0 and are not shown here.**

**Code and data availability.** The WRF-Chem model code is distributed by NCAR (https://doi.org/10.5065/D6MK6B4K, NCAR, 2020). The model data used to support the results described in this paper are available upon request to the first and corresponding authors. The synoptic data were downloaded from https://opendata.meteo.be/download, hosted by Royal Meteorological Institute (RMI). ICOS observations are available at https://meta.icos-cp.eu/collections/LKDae89cNpTOKSt1TnK_dRIw (ICOS RI et al., 2023). The TCCON data are available through the TCCON wiki at https://tccondata.org/. The ERA5 dataset is freely accessible after registration from the Copernicus Climate Data Store at https://cds.climate.copernicus.eu/datasets (Hersbach et al., 2020). CAMS global reanalyses, provided by the Copernicus Atmosphere Monitoring Service, were taken from https://ads.atmosphere.copernicus.eu/cdsapp#!/dataset/cams-global-reanalysis-eac4?tab=form (Inness et al., 2019). The CAMS-REG-ANT v8.0 emissions (Kuenen et al., 2022) and temporal profiles CAMS-REG-TEMPO v3.1 (Guevara et al., 2021) are archived and distributed through the Emissions of atmospheric Compounds and Compilation of Ancillary Data (ECCAD) platform. EDGAR emission inventory datasets are available at https://edgar.jrc.ec.europa.eu/dataset_ghg2024 (Crippa et al ., 2024). TNO_GHGco_v4.1 emission inventory was kindly provided by Ingrid Super.



**Author contributions.** JW and SCa designed the model setup. JW performed the analysis and wrote the paper. MZ and MDM provided general guidance and support during the analysis. SCo and MR provided the meteorological observations at OPE site. All authors reviewed and commented on the paper.

**Competing interests**. The contact author has declared that neither they nor their co-authors have any competing interests.

**Disclaimer.** Publisher's note: Copernicus Publications remains neutral with regard to jurisdictional claims made in the text, published maps, institutional affiliations, or any other geographical representation in this paper.

**Acknowledgements.** The authors would like to acknowledge the providers of emission inventories, as well as Emissions of Atmospheric Compounds and Compilation of Ancillary Data (ECCAD). We thank all members of the Synoptic observations,
TCCON, and ICOS Atmosphere Monitoring Station Assembly for providing the long-term and high-quality data on meteorological and greenhouse gas observations. Finally, we would like to thank Theo Glauch for his assistance in running PYVPRM.

**Financial support.** JW, SCa and this research have been supported by the National Key R&D Program (No. 2023YFC3705202), China Scholarship Council Program (Project ID: 202404910417), and the Belgian Research Action
through Interdisciplinary Networks PHASE 2 - 2018-2023 (BRAIN-be 2.0) project VERBE (contract n° B2/223/P1).

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
