# Peer review of "WRF-Chem simulations of CO2 over Western Europe assessed by ground-based measurements"

_EGUsphere, 2025_

## Referee Comment (RC2)

General comments:

This manuscript simulates $CO_2$ concentrations in Belgium and surrounding areas and conducts sensitivity tests using different emission inventories and in situ and remote-sensing $CO_2$ observations. The topic is interesting and meaningful, but many statements and explanations in the manuscripts are not rigorous enough. I suggest more modifications and improvements before acceptance.

Special comments:

1   The title includes Western Europe, but this study only focuses on Belgium and the surrounding areas. Is it reasonable to use Western Europe in the title?

2   This paper also mentioned that the drought could increase $CO_2$ concentrations. I think it is necessary to include precipitation in the research period and compare it with the year before and after.

3   I think Figures 1 and 2 can be combined.

4   Lines 130-135: I suggest consistency in the parameters used in the equation and the text. For example, "$T_{scale}$" in Eq and "Tscale" in text. Maybe "$T_{scale}$" is more suitable. Also, other parameters such as Wscale, Pscale, Ts, Tmin, Tmax, …

5   Lines 158-159, Here are two downscaling methods used. What are the differences between them?

6   Table 1. I suggest adding some words in the column of the aberrative. Also, the CBW attitude is 0?

7   Figures, the figures in the main text are not clear as the Figures in the Appendix.

8   Figure 3. What are the sunrise and sunset times in these ICOS stations? The highest temperature occurred at 18:00 local time, and the PBL reached its maximum height between 15:00-17:00 (line 319). It seems unreasonable.

9   Figure 5. I think it is better to keep the y-axis of $CO_2$ concentrations the same across different emission inventories at the same height.

10  Figure 6. It seems there is no contribution from biomass burning in these figures. Why are biogenic contributions nearly negative from 10:00- 23:00 at all sites? Why are biogenic contributions only positive around 10:00?

11  Figure 7. It is better to give the slope and correlation coefficient in Figure 7c for the three TCCON sites.

12  Figure 8. Although the STD and RMSE increase from S to P, MBE becomes large. Which parameters are more critical to evaluate the model's sensitivity?

13  Line 378. What does SNAP mean in this paper?

14  Figure 11. There is a large gap between SNAP 1 for CAMS and EDGAR. Figure 10d also showed more emission sources than Figure 1a. Why did this gap occur? Were the emissions included in other emission sectors in CAMS?

15  Figure 12. It seems that in late July and August, the land system was also active as a carbon source (Figure 12c), but anthropogenic emissions nearly disappeared from Figure 12b. Usually, drought can increase temperatures and the electricity demand for air conditioning, hence the anthropogenic emissions could increase in this period.

16  Lines 438-439, please add ° before N and E for the GPS location. Also, add this to the GPS location in Table A3. What does "acid fen" mean here for FR_LGt?

---

## Author Comment (AC1)

Dear Reviewer, we appreciate your time and effort in acknowledging and thoroughly reviewing our manuscript. We are sincerely grateful for your constructive comments and insightful suggestions, which have encouraged and helped us to improve the manuscript. We have revised the manuscript carefully based on your comments.

In the responses below, your comments are provided in black text and our responses are provided in blue text.

Wang et al. simulated $CO_2$ mole fractions over Western Europe in the summer of 2018 using the WRF-Chem model combined with three different $CO_2$ emission inventories. The simulations were evaluated by comparing with ground-based in situ and column observations. They showed, by taking into account the sector-specific vertical profiles of emissions, the agreement between the simulations and the observations was significantly improved for sites near large emission sources.

The topic of this manuscript is important and relevant to the scope of Atmospheric Chemistry and Physics. In addition, the analysis method is appropriate, and the writing structure is well organized. However, it seems to lack novelty and contains few new scientific findings. I recommend clarifying what is novel and addressing the following concerns and questions.

We thank the reviewer for the constructive comments. We summarize the main novel aspects of this study as the following two points, both of which have also been incorporated into the revised conclusion:

- This study provides the first high-resolution WRF-Chem simulation over Belgium and its surrounding countries, and evaluates the model performance using observations from multiple ICOS and TCCON sites. The results provide a systematic assessment of the applicability of the WRF-GHG model at regional scale in the core region of Western Europe, offering a solid basis for related modeling and applied studies.
- This study highlights the necessity of considering the vertical distribution of anthropogenic emissions of $CO_2$ in simulations, especially near strong emission sources.

Specific comments

1. L56: Please provide a clear description of whether the signal is positive or negative.

We appreciate the reviewer's suggestion.

The original sentence has been revised to: *Using the ICOS atmospheric measurements,*

*Ramonet et al. (2020) reported that a severe drought event in Europe in 2018 led to an atmospheric $CO_2$ signal of +1 to +2 ppm at most stations **in summer**.*

2.  L72: Do the multi-source observational data refer to ICOS and TCCON data? If so, I do not think they were used for simulating $CO_2$ mole fractions.

Yes, the multi-source observational data refer to ICOS and TCCON data. We added "and evaluate", and the sentence has been revised as: "*It employs the Weather Research and Forecasting Greenhouse Gas model (WRF-GHG; Beck et al., 2011) and multi-source observational data to simulate **and evaluate** $CO_2$ mole fractions over Western Europe, with a focus on Belgium and surrounding regions during summer 2018.*"

3.  L242–247: Please add a discussion of how the overestimation in inland areas and the underestimation in coastal regions affected the mean bias error in wind speed.

We appreciate the reviewer's suggestion. We have added some descriptions detailing the mean bias error (MBE) for inland and coastal stations. The corresponding sentence has been revised to: "*Besides, as shown in Fig. A1, the model tends to underestimate wind speed values significantly along coastal regions, with an MBE of -1.11 m/s across the four coastal stations, which is probably due to the coastal effects in the WRF simulation (Hahmann et al., 2015). In inland areas, the model tends to overestimate at most sites with dense vegetation cover. Such overestimation of wind speed has also been found in previous studies (Duan et al., 2018; Liu et al., 2022; Che et al., 2024). This bias may be attributed to the complex wind distribution in areas with rugged terrain, where the WRF model fails to adequately account for the additional resistance effects of vegetation on unresolved terrain, ultimately leading to an overestimation of wind speed. While an underestimation of wind speed is also observed at some inland stations, most of these stations are located at airports or in areas dominated by cropland. The land-use classification of the corresponding model grid cells is mostly identified as urban and built-up, which may lead to an overestimation of the surface roughness near the station and consequently an underestimation of wind speed. Similar findings were reported by Aylas et al. (2020), who found that WRF underestimated wind speed at airport stations by approximately -0.124 m/s, while the bias was reduced to -0.079 m/s after updating the land-use data.*" in lines 243-254 of the revised manuscript.

4.  L253: It would be easier to read if the expressions "between the observations and the simulations" and "between the simulations and the observations" were made consistent throughout this manuscript.

We appreciate the reviewer's suggestion. All related instances in the manuscript have

been uniformly revised to "*between simulations and observations*".

5.  Figures 8 and 9: The differences between the WRF-GHG simulations and TCCON data in Figure 9 appear to be larger than those in Figure 8. Are the differences due to the fact that Figure 9 does not take into account smoothing using the column averaging kernel?

Yes. In Figure 8 (i.e., Figure 7 in the revised manuscript), we use the simulated results after AVK smoothing. In Figure 9 (i.e., Figure 8 in the revised manuscript), since we can't apply smoothing to the background field and the individual tracers, all simulated results shown in Figure 9 are therefore presented without AVK smoothing in order to maintain consistency. The difference of $XCO_2$ before and after applying the AVK smoothing is around $1.08 \pm 0.44$ ppm (MBE ± STD) at the Orléans site, $1.18 \pm 0.34$ ppm at the Paris site, and $1.10 \pm 0.28$ ppm at the Karlsruhe site, respectively.

6.  338–347: If the data period for Orleans was matched with that for Paris, would similar results be obtained? In other words, would the difference between Paris (an urban area) and Orleans (a suburban area) be reflected in the observed $XCO_2$?

We thank the reviewer for the constructive comment.
We selected the data points where the observation times at the Orléans and Paris sites overlap, resulting in a sample number of 94. The statistical metrics between the simulated and observed $XCO_2$ at the Paris and Orléans site are given in the Table A, and the time series of observed and simulated $XCO_2$ using TNO inventory at Paris (magenta) and Orléans (teal) TCCON sites are shown in Figure A.
It can be found that, for this selected time series, compared to the large differences caused by using different emission inventories at the Paris site (up to approximately $0.49 \pm 0.33$ ppm between TNO and EDGAR), differences are also present at the Orléans site but are smaller (up to approximately $0.1 \pm 0.08$ ppm between TNO and CAMS). The simulated $XCO_2$ at the Orléans site (a suburban area) shows a systematic overestimation (around 0.4 - 0.5 ppm), but its magnitude is obviously lower than that at the Paris site (an urban area, around 1.1 - 1.6 ppm). As the anthropogenic emissions around Orléans are relatively low (Figure 9 in revised manuscript), it indirectly suggests that, in addition to uncertainties in the background fields, the overestimation at the Paris site is to a large extent caused by uncertainties in the anthropogenic emissions.

**Table A. The statistical metrics between the simulated $XCO_2$ using five different anthropogenic emission inputs and observed $XCO_2$ at Paris and Orléans sites. Here, the time series are selected based on the period overlapping between the Paris and Orléans site, resulting in N=94.**

| *Paris* |
| --- |

| | EDGAR_S | EDGAR_P | CAMS_S | CAMS_P | TNO_CAMS |
|---|---|---|---|---|---|
| MBE | 1.09 | 1.08 | 1.18 | 1.16 | 1.58 |
| STD | 0.84 | 0.84 | 0.87 | 0.86 | 0.96 |
| RMSE | 1.38 | 1.37 | 1.47 | 1.45 | 1.85 |
| R | 0.75 | 0.75 | 0.74 | 0.74 | 0.70 |
| **Orléans** | | | | | |
| MBE | 0.43 | 0.42 | 0.41 | 0.41 | 0.52 |
| STD | 0.76 | 0.76 | 0.77 | 0.77 | 0.77 |
| RMSE | 0.87 | 0.87 | 0.88 | 0.87 | 0.93 |
| R | 0.75 | 0.75 | 0.74 | 0.74 | 0.74 |

**Figure A. Time series (local time) of observed and simulated $XCO_2$ using TNO inventory at Paris and Orléans TCCON sites (b), and their absolute differences (WRF-GHG - TCCON) (a).**

7. L378: Please add an explanation of SNAP.

We appreciate the reviewer's suggestion. The sentence "*Anthropogenic sources are classified into 10 different categories according to the Selected Nomenclature for Air Pollutants (SNAP) in the study by Brunner et al. (2019) on vertical profiles.*" has been added to lines 169-170 of the revised manuscript.

8. L382–383: In Figure 10, it appears that there are also large emission sources near sites other than KIT (i.e., CBW and SAC). What degree of closeness does "near" represent?

Yes, in addition to the KIT site, there are also strong emission sources near the CBW and SAC sites. Figure B shows the simulated near surface $CO_2$ mole fractions at the lowest model level at each site, which is located at approximately 25 m above the ground. It can be found that at the TRN and OPE sites, the five simulated results are almost identical, which is consistent with the relatively weak anthropogenic emissions in their vicinity. At the SAC and CBW sites, whether vertical emissions are taken into

account indeed has an impact on the simulation results. Because the emission sources near the SAC site are comparatively weaker, the differences among the results there are also smaller. However, the sensitivity tests at the observation heights of SAC and CBW do not show significant biases. For SAC, this may be because the emission sources are not as strong. For CBW, this may be because the observation height used is relatively high (207 m above the ground). As shown by the results at different heights at the KIT site, the higher the observation height, the smaller the difference between simulations that consider only surface emissions and those that include vertical emissions.

[Figure]

**Figure B. Diurnal cycles (local time) of simulations with different anthropogenic emissions at five ICOS sites at the model lowest layer (approximately 25 m above the ground).**

For the degree of closeness represented by "near," since the influence of emission sources on a specific site is jointly affected by multiple factors, including wind direction, wind speed, emission source strength, and the distance to the site, we are sorry that we cannot provide a precise quantitative value.

We have added "*as well as near the CBW and SAC sites, whereas there are almost no emission sources in the vicinity of the TRN and OPE sites. Figure A4 shows the diurnal cycles of the five near-surface $CO_2$ mole fractions at each ICOS site as simulated by WRF-GHG fractions at the lowest model level (approximately 25 m above ground level). It can be found that at the TRN and OPE sites, the five simulations show nearly identical patterns, consistent with the relatively weak anthropogenic emissions in their*

*surrounding areas. In contrast, at the SAC and CBW sites, whether vertical emissions are taken into account does indeed affect the simulations. However, at the observation heights of these two sites, the impact is small and does not exhibit characteristics similar to those observed at the KIT site. For the SAC site, this may be due to the relatively weaker emission sources nearby, whereas for the CBW site, this may be related to the relatively high observation height used (207 m above ground level), as shown for the KIT site (Fig. 4), the discrepancies between sensitivity tests decrease with increasing observation height*" to lines 391-400 of the revised manuscript. In addition, Figure B has been incorporated into the revised manuscript as Figure A4.

9. L419: including -> includes

Done.

10. L431–434: The simulated $XCO_2$ values in July and August 2018 were higher than the observed $XCO_2$. What caused this overestimation by the model? In years other than 2018 when no drought occurred, will the simulated $XCO_2$ values be lower than the observed values?

This overestimation may be due to an overestimation of anthropogenic emissions, or an overestimation of biogenic fluxes, or a combination of both effects.

As our present study only focuses on simulations conducted for the period from June to August 2018, we are unfortunately unable to provide simulation results outside this time window. Therefore, we regret that we cannot currently elaborate on how the simulated $XCO_2$ would behave in years without drought conditions. Nevertheless, we appreciate the reviewer's insightful comment.

11. L454: analyze -> analyzing

Done.

12. L486–488: Please revise the sentence by adding a conjunction.

We appreciate the reviewer's suggestion. We have added the conjunction "as" to the sentence. The revised sentence is as follows: *At the same time, observational data play a critical role in model evaluation, **as** the availability of time-synchronized ground-based observations of $CO_2$ fluxes and concentrations can help assess the performance of the model, thereby enhancing the credibility and scientific interpretation of the simulation results.*

**Reference**

Aylas, Y.G.R., De Souza Campos Correa, W., Santiago, A.M., Reis Junior, N.C., Albuquerque, T.T.A., Santos, J.M., Moreira, D.M.: Influence of land use on the performance of the WRF model in a humid tropical climate, Theor. Appl. Climatol., 141, 201–214, https://doi.org/10.1007/s00704-020-03187-3, 2020.

Aylas, Y.G.R., De Souza Campos Correa, W., Santiago, A.M., Reis Junior, N.C., Albuquerque, T.T.A., Santos, J.M., Moreira, D.M.: Influence of land use on the performance of the WRF model in a humid tropical climate, Theor. Appl. Climatol., 141, 201–214, https://doi.org/10.1007/s00704-020-03187-3, 2020.

---

## Author Comment (AC2)

Dear Reviewer, we appreciate your time and effort in acknowledging and thoroughly reviewing our manuscript. We are sincerely grateful for your constructive comments and insightful suggestions, which have encouraged and helped us to improve the manuscript. We have revised the manuscript carefully based on your comments.

In the responses below, your comments are provided in black text and our responses are provided in blue text.

General comments:

This manuscript simulates $CO_2$ concentrations in Belgium and surrounding areas and conducts sensitivity tests using different emission inventories and in situ and remote-sensing $CO_2$ observations. The topic is interesting and meaningful, but many statements and explanations in the manuscripts are not rigorous enough. I suggest more modifications and improvements before acceptance.

Special comments:

1.   The title includes Western Europe, but this study only focuses on Belgium and the surrounding areas. Is it reasonable to use Western Europe in the title?

We appreciate the reviewer's suggestion. The title has been revised as "*WRF-Chem simulations of $CO_2$ over Belgium and surrounding countries assessed by ground-based measurements*".

2.   This paper also mentioned that the drought could increase $CO_2$ I think it is necessary to include precipitation in the research period and compare it with the year before and after.

We appreciate the reviewer's suggestion. The sentence "*According to records from the Royal Meteorological Institute of Belgium (RMI) at the Uccle station, the total precipitation for the summer (June to August) of 2018 was 134.7 mm, which was substantially lower than the climatological mean for the summer period 1991–2020 (234.2 mm).*" has been added to lines 101-103 of the revised manuscript.

3.   I think Figures 1 and 2 can be combined.

Done. The figure has been revised as follows.

[Figure]

*Figure 1. Terrain elevation map of the simulated domains, with horizontal resolutions of 9 km (d01) and 3 km (d02), showing ICOS (yellow dots), TCCON (orange diamond) and co-located (red stars) sites within d02 (a), and synoptic stations in Belgium for which data are available for our study period (b).*

4.   Lines 130-135: I suggest consistency in the parameters used in the equation and the text. For example, "$T_{scale}$" in Eq and "Tscale" in text. Maybe "$T_{scale}$" is more suitable. Also, other parameters such as Wscale, Pscale, Ts, Tmin, Tmax, …

Done.

5.   Lines 158-159, Here are two downscaling methods used. What are the differences between them?

In these two methods, the applied time factors are different.

For the CAMS-REG-ANT emission inventory, we applied the temporal factors provided by CAMS-REG-TEMPO for downscaling, as this dataset is consistent with CAMS-REG-ANT in terms of sector classification, spatial resolution, and geographical coverage. In this method, sector-specific temporal factors were used for the downscaling. For example, emissions from the sector H_Aviation were assumed to have no temporal variability, therefore, the annual emissions were simply divided by the total number of hours in the year to obtain hourly emissions. While emissions from the sector A_PublicPower vary by grid cell on monthly, weekly (day-of-week), and hourly timescales, and these corresponding temporal factors were applied to downscale the annual emissions from the A_PublicPower sector. Similarly, for the remaining sectors, we also applied their respective temporal factors. Finally, emissions from each sector,

after applying the corresponding temporal factors, were summed to obtain the total hourly emissions.

For the TNO emission inventory, we used the downscaling method proposed by Nassar et al. (2013). This method downscales the total anthropogenic emissions instead of applying sector-specific downscaling. In this method, anthropogenic emissions vary by grid cell according to the day of the week (Monday to Sunday) and on hourly timescales, after being interpolated onto the WRF grid, these time factors are used to downscale the total anthropogenic emissions.

6.   Table 1. I suggest adding some words in the column of the aberrative. Also, the CBW attitude is 0?

In Table 1, it seems that we did not address the term "aberrative."

The CBW site is located in the Netherlands, and its elevation is 0 m; this information is obtained from the ICOS website.

7.   Figures, the figures in the main text are not clear as the Figures in the Appendix.

We apologize for the inconvenience. All figures included in the manuscript comply with the ACP journal requirements and are provided in PNG format with a resolution of 300 dpi. Some figures may appear less clear because they contain a large amount of information and have large original sizes, which may reduce visual clarity when the figures are scaled. We have revised most of the figures in the main text to improve their clarity as much as possible.

8.   Figure 3. What are the sunrise and sunset times in these ICOS stations? The highest temperature occurred at 18:00 local time, and the PBL reached its maximum height between 15:00-17:00 (line 319). It seems unreasonable.

We thank the reviewer for this comment. During the simulation period, according to NOAA Solar Calculator, the sunrise at the ICOS sites involved in our study occurs around 5:00 - 7:00 (local time), and sunset occurs around 20:00 - 22:00, and the sun reaches its highest position around 14:00 LT at most sites (energy saving time, UTC+2).

We have double-checked. As shown in Figure 2 of the revised manuscript, the simulated diurnal variation of temperature is consistent with observations. For the PBL, we extracted the corresponding PBL data from ERA5. Figure A presents the diurnal cycles of the PBL provided by ERA5 (blue) at each ICOS site. It can be seen that, at all sites, the diurnal variation of the PBL simulated by the WRF model is generally consistent with that from ERA5. Therefore, the consistency with observations and ERA5 data

demonstrates the reliability of the results.

[Figure]

**Figure A. The diurnal cycles (local time, UTC+2) of planetary boundary layer height simulated by WRF-GHG (red) and ERA5 (blue) at the KIT (a), the TRN (b), the SAC (c), the OPE (d) and the CBW (e) sites.**

9. Figure 5. I think it is better to keep the y-axis of $CO_2$ concentrations the same across different emission inventories at the same height.

Done. The figure has been revised as follows.

[Figure]

*Figure 4. Diurnal cycles (local time) of simulations with different anthropogenic emissions and observations (a-h), where the values represent the MBE between each simulation and observations, along with scatterplots comparing each simulation to the observations (i-l) at different heights at the ICOS KIT site.*

10. Figure 6. It seems there is no contribution from biomass burning in these figures. Why are biogenic contributions nearly negative from 10:00- 23:00 at all sites? Why are biogenic contributions only positive around 10:00?

The contributions from biomass burning and the ocean are both nearly zero; therefore, in Figure 6 (Figure 5 in revised manuscript), the biomass burning values are overlapped by the ocean contribution, making them appear not to be shown.

[Figure]

**Figure B. Diurnal cycles of biogenic fluxes (GPP, RES, and Total) at five ICOS sites.**

[Figure]

*Figure 5. Diurnal cycles (local time) of simulated tracer contributions at 30 m at the KIT (a), 50 m at the TRN (b), 60 m at the SAC (c), 50 m at the OPE (d) and 207 m at the CBW (e) sites. Here, the anthropogenic emissions are based on EDGAR v2024, taking into account the vertical emission profiles.*

[Figure]

*Figure A3. The diurnal cycles (local time) of planetary boundary layer height simulated by WRF-GHG at the KIT (a), the TRN (b), the SAC (c), the OPE (d) and the CBW (e) sites.*

Across all sites, during the local time period of approximately 10:00 - 23:00, the biogenic contribution is nearly negative. This behavior is primarily attributed to the fact that daytime $CO_2$ uptake by vegetation through photosynthesis is substantially stronger than ecosystem respiration. As shown in the diurnal cycle of biogenic fluxes (Figure B), gross primary productivity (GPP) increases rapidly after sunrise (05:00 - 07:00) and acts as a strong negative flux. Its magnitude clearly exceeds that of respiration (RES) during the daytime, resulting in a net negative biogenic $CO_2$ flux, which indicates that terrestrial ecosystems act as a net sink of atmospheric $CO_2$. The period from sunrise to approximately 10:00 corresponds to the morning transition phase. During this period, photosynthesis gradually intensifies while respiration continues. Due to $CO_2$ accumulation caused by nighttime respiration under shallow PBL conditions, the biogenic contribution to near-surface $CO_2$ concentrations during this period remains positive. However, as photosynthesis strengthens, the overall biogenic contribution to near-surface $CO_2$ concentrations exhibits a decreasing trend.

Around 10:00 - 11:00, photosynthesis is strong at all sites, and the total biogenic flux reaches a strong $CO_2$ uptake (Figure B), and simultaneously, the PBL height starts rising rapidly (Figure A3). At this time, photosynthetic $CO_2$ uptake largely offsets the $CO_2$ accumulated from nighttime respiration. Subsequently, as the biogenic flux remains in a net uptake regime throughout the daytime, the biogenic contribution to near-surface $CO_2$ concentrations remains stably negative from daytime until approximately 20:00.

After 20:00, as photosynthesis weakens and ecosystem respiration becomes dominant again, the total biogenic flux gradually shifts to positive values. Meanwhile, the height of PBL decreases, facilitating the accumulation of respiration-derived $CO_2$ near the surface. Consequently, the biogenic contribution to near-surface $CO_2$ concentrations transitions from negative values toward zero and largely compensates for the daytime uptake by around 23:00.

We apologize if we may not have fully understood the reviewer's question. We infer that the last question may be: *Why are the biogenic contributions **at the CBW site** only positive around 10:00, whereas other sites show a different pattern?* For the CBW site, the biogenic contribution is only positive around 9:00-10:00, in contrast to the other sites where positive values most persist from 01:00 to 10:00. This difference may be attributed to the higher observation height at the CBW site (207 m above ground), whereas the observation heights at the other sites are around 50 m. At night, $CO_2$ released by respiration remains difficult to transport to the observation height of 207m because the nocturnal PBL is shallow and the atmospheric stratification is relatively stable (as shown in Figure A3(e), where the height of PBL at night is mostly below 200m). The positive value around 9:00-10:00 is likely caused by the onset of vertical mixing and the rise of PBL height, which transports $CO_2$ accumulated within the nocturnal boundary layer upward to the observation height. Before 9:00, the observation height is decoupled from the nocturnal boundary layer and is not influenced by nighttime accumulation and respiration, but rather by the remainder of the daytime uptake (residual layer), resulting in negative concentrations. After 10:00, as the PBL gradually deepens during the daytime and vertical mixing is enhanced, near-surface $CO_2$ is transported to higher altitudes. However, during this period, photosynthetic $CO_2$ uptake dominates, and thus the observed $CO_2$ concentration changes are predominantly negative.

11. Figure 7. It is better to give the slope and correlation coefficient in Figure 7c for the three TCCON sites.

We appreciate the reviewer's suggestion. We have revised the figure as follows.

[Figure]

*Figure 6. Time series (local time) of simulated XCO2 using TNO inventory and observed XCO2 at three TCCON sites (b), their absolute differences (WRF-GHG - TCCON) (a), and scatterplots (c). "N" represents the number of data pairs, "a" represents the slope of the linear regression curve and "R" represents the correlation coefficient.*

12. Figure 8. Although the STD and RMSE increase from S to P, MBE becomes large. Which parameters are more critical to evaluate the model's sensitivity?

We thank the reviewer for this insightful comment. Indeed, at the KIT site, the MBE becomes larger when vertical emissions are considered (P) compared to the only surface emissions (S). However, the uncertainty associated with the TCCON $XCO_2$ measurements is approximately 0.5 ppm. The difference in MBE remains within the observational uncertainty.

Regarding the four statistical metrics (MBE, STD, RMSE, and R), we acknowledge that it's difficult to identify one as the most critical for evaluating model sensitivity. Each metric highlights different aspects of model performance, these metrics are generally interpreted together rather than individually. In this study, we find that except for MBE, the other three metrics exhibit slight improvements when vertical emissions are considered. Given that the change in MBE remains within the $XCO_2$ uncertainty range, we conclude that the overall improvement in $XCO_2$ simulation is present but relatively small. This is why we state in the manuscript that *"we observe a small improvement in the simulation of XCO₂, but not as notable as that observed for near-surface mole fractions."*

13. Line 378. What does SNAP mean in this paper?

SNAP stands for Selected Nomenclature for Air Pollutants. Following this classification, anthropogenic emissions are provided separately for different source categories. The sentence "*Anthropogenic sources are classified into 10 different categories according to the Selected Nomenclature for Air Pollutants (SNAP) in the study by Brunner et al. (2019) on vertical profiles.*" has been added to lines 169-170 of the revised manuscript.

14. Figure 11. There is a large gap between SNAP 1 for CAMS and EDGAR. Figure

10d also showed more emission sources than Figure 1a. Why did this gap occur? Were the emissions included in other emission sectors in CAMS?

The sectoral classifications of anthropogenic emissions differ between CAMS-REG-ANT and EDGAR. In CAMS-REG-ANT, anthropogenic emissions are categorized into 12 sectors (A - L) according to Gridded Nomenclature For Reporting (GNFR) categories, whereas EDGAR v2024 classifies anthropogenic emissions into 8 sectors.

In order to be able to apply vertical profiles to them, we mapped them to sectors classified according to the Selected Nomenclature for Air Pollutants (SNAP), which comprises a total of 10 sectors. However, due to the inconsistent criteria used for sectoral allocation in EDGAR and CAMS-REG-ANT, there may be some degree of cross-sectoral overlap between these datasets during the mapping process.

Besides, we also plotted the maps of the total annual anthropogenic emissions from CAMS-REG-ANT and EDGAR v2024 for 2018 (see Figure C), respectively. It can be seen that the annual emissions provided by EDGAR are significantly higher than those from CAMS.

[Figure]

**Figure C. Map of total anthropogenic emissions from CAMS-REG-ANT and EDGAR v2024 in 2018.**

15. Figure 12. It seems that in late July and August, the land system was also active as a carbon source (Figure 12c), but anthropogenic emissions nearly disappeared from Figure 12b. Usually, drought can increase temperatures and the electricity demand for air conditioning, hence the anthropogenic emissions could increase in this period.

We thank the reviewer for this comment. Indeed, there is a carbon source near the surface in July and August for the biogenic tracer which can be seen in Fig. 12c (Figure 11 in revised manuscript), but this feature is less pronounced in Fig. 12b since the latter represents the column-averaged dry-air mole fractions of $CO_2$. As shown in Fig. 12c, the contribution of biogenic $CO_2$ decreases with increasing altitude and becomes negligible above approximately 2000 m.

And indeed, drought could increase electricity demand, but air conditioning remains relatively uncommon in Europe, especially in the countries involved in our study. In addition, Figure D shows the time series of monthly anthropogenic emissions from EDGAR v2024 obtained from ECCAD at the Paris (a), Karlsruhe (b), and Orléans (c) sites, respectively. The shaded area represents emissions during June - August of each year. Compared with non-drought years (e.g., 2016 and 2017), anthropogenic emissions in summer 2018 do not exhibit a noticeable increase and are even slightly lower than in those years. Based on these two considerations, the lack of a noticeable increase in simulated anthropogenic emissions during this period is reasonable.

[Figure]

**Figure D. Time series of monthly anthropogenic emissions from EDGAR v2024 at the Paris (a), Karlsruhe (b), and Orléans (c) sites.**

16. Lines 438-439, please add ° before N and E for the GPS location. Also, add this to the GPS location in Table A3. What does "acid fen" mean here for FR_LGt?

We appreciate the reviewer's suggestion. We have added ° at the corresponding locations in the manuscript.

Here the "acid fen" refers to a type of wetland characterized by acidic conditions, typically with low pH levels. The ICOS description of this site is as follows: "*The La Guette station (FR-LGt) is a peatland located in Neuvy sur Barangeon (Sologne) at about 200 Km south of Paris and 80 Km south of the Université d"Orléans. It is an acid fen that is cut at the output by a road.*"